# Structural and functional determination of peptide versus small molecule ligand binding at the apelin receptor

Thomas L. Williams [1,7], Grégory Verdon [2,7], Rhoda E. Kuc[1], Heather Currinn[2], Brian Bender[2], Nicolae Solcan[2], Oliver Schlenker[2], Robyn G. C. Macrae[1,3], Jason Brown[2], Marco Schütz[2], Andrei Zhukov[2], Sanjay Sinha [3], Chris de Graaf[2,8], Stefan Gräf [4,5,6,8], Janet J. Maguire [1,8], Alastair J. H. Brown[2,8] ✉ & Anthony P. Davenport [1,8] ✉

We describe a structural and functional study of the G protein-coupled apelin receptor, which binds two endogenous peptide ligands, apelin and Elabela/Toddler (ELA), to regulate cardiovascular development and function. Characterisation of naturally occurring apelin receptor variants from the UK Genomics England 100,000 Genomes Project, and AlphaFold2 modelling, identifies T89[2.64] as important in the ELA binding site, and R168[4.64] as forming extensive interactions with the C-termini of both peptides. Base editing to introduce an R/H168[4.64] variant into human stem cell-derived cardiomyocytes demonstrates that this residue is critical for receptor binding and function. Additionally, we present an apelin receptor crystal structure bound to the G protein-biased, small molecule agonist, CMF-019, which reveals a deeper binding mode versus the endogenous peptides at lipophilic pockets between transmembrane helices associated with GPCR activation. Overall, the data provide proof-of-principle for using genetic variation to identify key sites regulating receptor-ligand engagement.

The apelin receptor belongs to the G protein-coupled receptor (GPCR) superfamily, comprising nearly 800 transmembrane proteins that provide targets for over a third of all FDA-approved medicines[1–4].

The receptor binds two endogenous peptide ligands, apelin and Elabela/Toddler (ELA), proteolytically cleaved from preproproteins to yield smaller, active fragments such as [Pyr[1]]apelin-13 and ELA-11 (Fig. 1a)[1,5–11]. The existence of two different ligands is a distinct feature of apelin receptor biology, with the peptides sharing little sequence identity but binding with similar affinities and both being blocked by antagonists[7,8,10]. In developmental physiology, the apelin receptor is critical for transducing predominantly ELA signalling to drive proper formation of the heart during early embryogenesis. This points to temporal differences between apelin and ELA, although further physiological consequences for having two peptides remain unclear[7,8,10].

The in vitro[6–12] and in vivo[7,9–11,13–17] cardiovascular actions of apelin and ELA have been extensively characterised. In the adult cardiovascular system, both apelin and ELA importantly induce antihypertensive effects, mediating venous and arterial vasodilatation[6,10–15]. In the heart, apelin induces potent positive inotropy, increasing cardiac contractility and output through activation of apelin receptors present on

[1]Experimental Medicine & Immunotherapeutics, University of Cambridge, Cambridge, UK. [2]Nxera Pharma UK Limited (Sosei Heptares), Steinmetz Building, Granta Park, Cambridge, UK. [3]Wellcome-MRC Cambridge Stem Cell Institute, Jeffrey Cheah Biomedical Centre, University of Cambridge, Cambridge, UK. [4]NIHR BioResource for Translational Research – Rare Diseases, Cambridge Biomedical Campus, Cambridge, UK. [5]Department of Haematology, NHS Blood and Transplant, Long Road, University of Cambridge, Cambridge, UK. [6]Department of Medicine, University of Cambridge, Victor Phillip Dahdaleh Heart & Lung Research Institute, Cambridge, UK. [7]These authors contributed equally: Thomas L. Williams, Grégory Verdon. [8]These authors jointly supervised this work: Chris de Graaf, Stefan Gräf, Janet J. Maguire, Alastair J. H. Brown, Anthony P. Davenport. ✉e-mail: Alastair.Brown@Nxera.life; apd10@medschl.cam.ac.uk

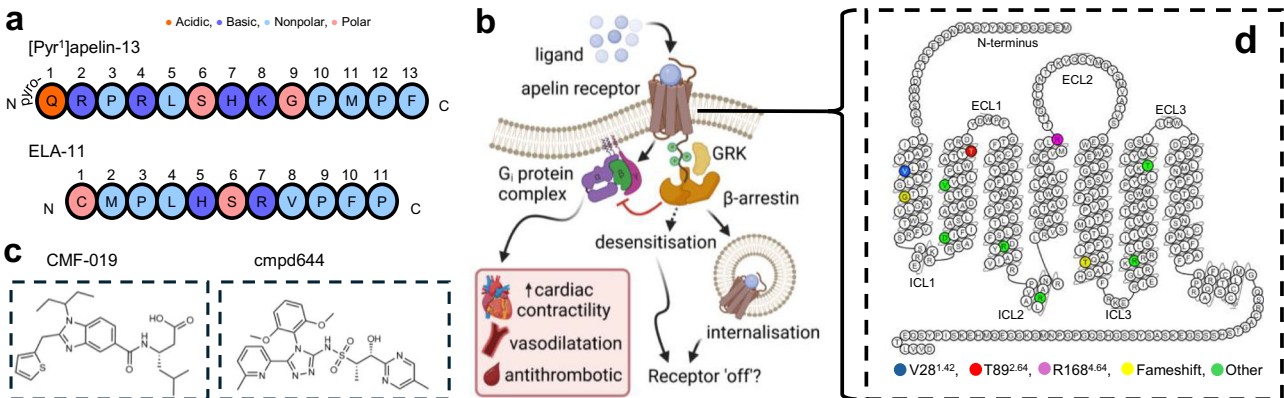

**Fig. 1 | Overview of apelin receptor ligands that regulate the cardiovascular system in health and disease. a** Aligned amino acid sequences of the endogenous apelin receptor peptides, [Pyr¹]apelin-13 (upper) and ELA-11 (lower). Amino acids are provided as single letter codes and coloured according to classification, as indicated in figure. **b** Schematic showing the signalling cascades following ligand binding and activation of apelin receptor present on the cell surface. Canonical $G_i$ protein signalling mediates physiological effects of the apelin receptor, including increasing cardiac contractility, inducing vasodilatation, and antithrombotic action. Prolonged binding of an agonist can facilitate phosphorylation of the receptor by G protein-coupled receptor kinase (GRK), and subsequent recruitment of β-arrestin protein. This typically desensitises and 'switches off' receptor signalling by sterically hindering engagement with G protein complexes and inducing internalisation of the receptor. Created in BioRender. Davenport, A. (2024) https://BioRender.com/f02c819. **c** Chemical structures of two apelin receptor small molecule agonists, CMF-019 (left) and cmpd644 (right). The structures show differences in chemotypes between the two ligands. **d** Snake plot of the apelin receptor showing the 380 amino acid sequence generated using GPCRdb (https://gpcrdb.org/)[37]. Colours indicate the residue locations for missense (V38, blue; T89, red; R168, magenta) or frameshift (G45, T227, both yellow) variants identified in individuals recruited to the 100,000 Genomes Project. Six other missense variants (green) were identified but were not further characterised in vitro, as they did not significantly alter cell surface expression or [¹²⁵I]-apelin-13 radioligand binding compared to wild-type.

cardiomyocytes[13,14,16–19]. Apelin signalling also plays a role in haemostasis, exerting antithrombotic effects through activation of apelin receptors present on platelets[20]. Beyond the cardiovascular system, the apelin signalling axis is widely distributed in peripheral and central tissues[11]. Dysregulation of apelin receptor signalling can, therefore, contribute to a range of disease states, and there is much interest in the apelin receptor as a therapeutic target in cardiovascular, metabolic, immune, renal, and neurological conditions[9,11].

Downregulation of apelin signalling in pulmonary arterial hypertension has gathered significant interest, and targeting the apelin receptor with agonists forms a future potential therapeutic strategy[10,21–24]. However, whilst exogenous peptide agonists are widely used as therapeutic agents, they typically exhibit poor absorption and stability in vivo, severely affecting receptor engagement and, therefore, efficacy[25]. Furthermore, treatment with agonists can be associated with secondary processes such as receptor desensitisation, typically in the form of internalisation, which can limit their effectiveness (Fig. 1b)[26]. Several distinct small molecule agonists targeting the apelin receptor have been developed in recent years, including AMG 986[27,28], cmpd644[29], and the G protein-biased agonist CMF-019 (Fig. 1c), which triggers less receptor internalisation and desensitisation, providing an interesting profile worth further examination, notably around structure activity relationships at the apelin receptor[30].

Previously, experimental structures of the apelin receptor have been reported for complexes with the synthetic peptide AMG3054[31], a synthetic single-domain antagonist antibody (JN241)[32], the endogenous peptide ELA and small synthetic agonist cmpd644[29], and more recently with the endogenous peptide apelin and the G protein-biased small synthetic agonist CMF-019 by cryoEM[33]. While these structures have provided insights into receptor binding modes, key determinants of engagement with endogenous peptides versus small synthetic agonists remain largely uncharacterised.

To provide further clarity on peptide versus small molecule binding at the apelin receptor, we used a combinatorial approach. The UK Genomics England 100,000 Genomes Project was developed to carry out the sequencing of 100,000 whole genomes from National Health Service patients with rare diseases and cancers[34], and provides the opportunity to identify naturally occurring variants within functional regions, including endogenous ligand and drug binding sites, of protein targets such as GPCRs[35,36]. To our knowledge, naturally occurring human apelin receptor variants have not been investigated, so here we explored several rare variants (Fig. 1d, Table 1) identified in the 100,000 Genomes Project that were predicted to be deleterious to receptor function in an in vitro setting. AlphaFold2 modelling was then used to propose a structural rationale for the impact of receptor variants on ligand binding. Additionally, we leveraged Nxera Pharma (formerly Sosei Heptares) proprietary technology to generate a stabilised receptor (NxStaR®), enabling the determination of an apelin receptor crystal structure in complex with the G protein-biased, small molecule agonist CMF-019[30].

In this work, our structure showed an overall pose of CMF-019 comparable with that reported recently[33], but with some notable differences. Overall, CMF-019 bound deeper into the orthosteric site than peptide ligands and engaged with lipophilic pockets between transmembrane helices associated with GPCR activation. Our functional, modelling, and structural data highlight the key determinants of endogenous peptide and small synthetic agonist engagement at the apelin receptor, including at naturally occurring coding variants that show differentiated binding modalities. Importantly, our study identifies and further characterises key regions and residues of the receptor involved in small molecule agonist binding, to inform future drug discovery research.

## Results

### Identification and pharmacological characterisation of apelin receptor variants

As part of our strategy to identify key residues involved in apelin receptor binding and function, we selected eleven ultra-rare (i.e. allele frequency <1 in 10,000), naturally occurring variants from individuals recruited to the NIHR BioResource (NBR) component of the 100,000 Genomes Project. Predetermined exclusion criteria (see Methods) aimed to ensure that selected variants did not occur at known or predicted post-translational modifications (e.g. glycosylation, palmitoylation, or disulfide bridges) or phosphorylation sites,

**Table 1 | Summary of ultra-rare apelin receptor variants identified in individuals recruited to the 100,000 Genomes Project NBR**

| Variant | Protein location | Genomic coordinates (chr:position) | Variant type | HGVS coding sequence | Codon change | HGVS protein sequence | SIFT | Poly Phen-2 |
|---|---|---|---|---|---|---|---|---|
| V/L38[1.42]* | TM1 | 11:57004367 | missense | 112G>C | GTC →CTC | V38L | 0.02 | 0.931 |
| G/X45[1.49]* | TM1 | 11:57004344 | frameshift | 134delG | GGC→GCΔ | G45X | n/a | n/a |
| D/V65[2.40] | TM2 | 11:57004285 | missense | 194A>T | GAT→GTT | D65V | 0.00 | 0.999 |
| V/M79[2.54] | TM2 | 11:57004244 | missense | 235G>A | GTG→ATG | V79M | 0.00 | 1.000 |
| T/M89[2.64]* | TM2 | 11:57004213 | missense | 266C>T | ACG→ATG | T89M | 0.00 | 1.000 |
| R/H127[3.50] | TM3 | 11:57004099 | missense | 380G>A | CGC→CAC | R127H | 0.00 | 1.000 |
| R/W139[34.55] | ICL2 | 11:57004064 | missense | 415C>T | CGG→TGG | R139W | 0.00 | 1.000 |
| R/H168[4.64]* | TM4 | 11:57003976 | missense | 503G>A | CGC→CAC | R168H | 0.00 | 1.000 |
| T/X227[5.64]* | TM5 | 11:57003799 | frameshift | 679delA | ACC→ΔCC | T227X | n/a | n/a |
| R/W243[6.30] | TM6 | 11:57003752 | missense | 727C>T | CGG→TGG | R243W | 0.00 | 1.000 |
| T/M269[6.56] | TM6 | 11:57003673 | missense | 806C>T | ACG→ATG | T269M | 0.00 | 1.000 |

The table summarises the variants, providing the residue sequence location and Ballesteros–Weinstein numbering, genomic co-ordinates (chromosome#:position#), the Human Genome Variation Society (HGVS) coding and apelin receptor sequence mutations codon changes. Sorting Intolerant From Tolerant (SIFT) and PolyPhen-2 scores indicate predicted deleteriousness. For SIFT, the amino acid substitution is predicted damaging if the score is ≤0.05, and tolerated if the score is >0.05. For PolyPhen-2, values closer to 1.0 are more confidently predicted to be deleterious. * in the first column indicates variants that were subsequently characterised in vitro. TM = transmembrane domain; ICL = intracellular loop; chr = chromosome; Δ = nucleotide deletion in codon. Colours indicate the missense (V/L38[1.42], blue; T/M89[2.64], red; R/H168[4.64], magenta) or frameshift (G/X45[1.49], T/X227[5.64], both yellow) variants. Six other missense variants (green) were also identified but were not further characterised in vitro.

and were predicted to be deleterious to protein structure and/or function.

Two variants, G/X45[1.49] and T/X227[5.64], comprised single nucleotide deletion frameshift mutations, and the other nine harboured missense mutations. Of the nine missense mutations, V/L38[1.42], T/M89[2.64], and R/H168[4.64] were selected for in-depth pharmacological characterisation based on a preliminary binding screen and/or a current lack of experimental knowledge for these residues. Further details on the apelin receptor variants are outlined in Table 1.

To characterise apelin receptor variants in vitro, CHO-K1 cells were transiently transfected with constructs encoding variant or wild-type receptor tagged with an enhanced green fluorescent protein (eGFP) reporter at the C-terminus. High content screening was used to qualitatively and quantitatively assess expression and fluorescent ligand binding in whole cells using the recently characterised apelin647 and ELA647 fluorescently labelled peptides[38]. Radioligand binding using [125I]-apelin-13 was also performed to empirically determine specific binding at receptor variants. A schematic of the experimental pipeline is provided in Fig. 2a, and a summary of numerical pharmacological data are provided in Table 2.

Our analyses showed that the missense variants, V/L38[1.42], T/M89[2.64], and R/H168[4.64] were expressed and localised to the cell membrane similarly to the wild-type receptor (Fig. 2b–d). However, the two variants, G/X45[1.49] and T/X227[5.64], were expressed at lower levels at the cell membrane, and were expressed in fewer cells overall than wild-type receptor (Fig. 2b–d).

Apelin647 and ELA647 fluorescent ligand both bound to wild-type receptor at the cell surface (Fig. 2b, e). As might be expected, fluorescent ligand binding was significantly lower than wild-type for both frameshift variants, likely due to a combination of reduced membrane expression and improper protein folding. The missense V/L38[1.42] variant bound both apelin647 and ELA647 in a manner similar to wild-type. Intriguingly, making a distinction between the two endogenous peptide isoforms, T/M89[2.64] bound apelin647, but ELA647 fluorescence was significantly lower at this variant versus the wild-type receptor. Finally, R/H168[4.64] showed very little binding with either fluorescent ligand.

[125I]-apelin-13 radioligand binding confirmed loss of ligand engagement at the G/X45[1.49] and T/X227[5.64] frameshift variants (Table 2). Interestingly, we were also unable to detect binding at the V/L38[1.42] variant (Fig. 2f, Table 2), despite the fact that we had previously observed binding of apelin647 and ELA647. Binding was

observed for T/M89[2.64], which displayed a $K_D$ similar to wild-type (Fig. 2f, Table 2). Further radioligand competition binding studies showed that the T/M89[2.64] variant required nearly 30-fold higher concentrations of ELA-11 to displace the radiolabel versus wild-type (Fig. 2g), and, overall, our in vitro data suggest that this residue site more tightly regulates engagement with ELA peptides than apelin. In line with the fluorescent ligand data, R/H168[4.64] was unable to bind [125I]-apelin-13 (Fig. 2f, Table 2), suggesting that this residue is directly involved in regulating endogenous peptide engagement.

Functional assessment of the apelin receptor variants in response to endogenous [Pyr1]apelin-13 was performed using a dynamic mass redistribution assay in transfected CHO-K1 cells (Table 2 and Fig. S1). Functional responses were observed for wild-type, V/L38[1.42], and T/M89[2.64] variants. However, consistent with the absence of receptor-ligand binding, R/H168[4.64] was non-functional in response to [Pyr1]apelin-13.

## AlphaFold2 models provide structural insights into peptide binding

To understand the potential structural impact of the T/M89[2.64] and R/H168[4.64] variants on apelin and ELA binding, we analysed peptide binding modes in previously determined peptide-bound structures. We also used AlphaFold2 to generate models of the apelin receptor in an active mini-$G_i$ bound conformation and bound to either apelin or ELA (Fig. 3).

Recently, AlphaFold2 has been shown to predict structures of proteins, including GPCRs and peptide binding modes, providing a rationale to understand the possible effects of mutations on structure and binding[39,40]. Overall, our apelin receptor models were very similar to previous peptide-bound structures and consistently showed the proximity of the T89[2.64] and R168[4.64] residues to the C-termini of the bound peptides near the bottom of the orthosteric site (Figs. 3a, and S2). Based on its position below the orthosteric site, we speculate that the V/L38[1.42] substitution triggers side chain movement towards peptide C-termini, potentially clashing with the C-terminally labelled tyrosine of the [125I]-apelin-13 radiolabel, but not with [Pyr1]apelin-13, or N-terminally labelled apelin647, or ELA647 fluorescent ligands. This might explain why we only observed loss of binding of the radioligand at this variant in in vitro experiments (Fig. 2).

While the R/H168[4.64] mutation could impact the local conformation of the receptor around the junction between TM4 and ECL2, R168[4.64] is involved directly in peptide binding, with its guanidinium

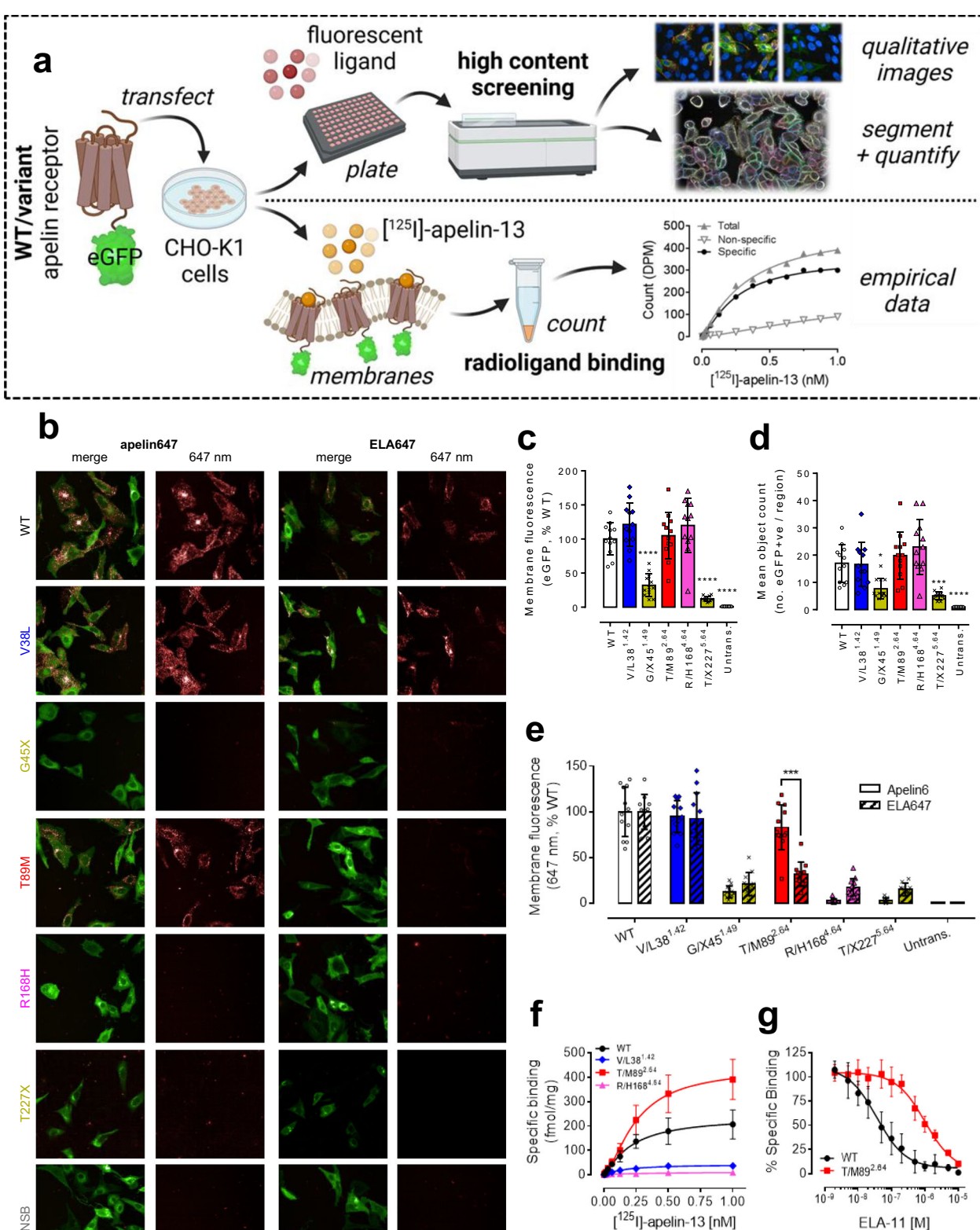

group interacting with peptide backbone oxygens in the AMG3054 peptide[31], the synthetic antibody JN241[32], and ELA in one of the recent cryoEM structures[29] (Fig. S2). In our AlphaFold2 models of complexes with apelin and ELA, the R168[4.64] guanidinium group falls within hydrogen bonding distance of the peptide C-terminus, strongly interacting with the last carbonyl and the terminal carboxyl (Fig. 3b

Fig. S3A–C), further emphasising R168[4.64] as a key determinant of apelin receptor binding to both endogenous peptides.

Modelling of the T/M89[2.64] variant suggests that the methionine substitution would narrow the binding pocket for the ELA peptide and create a steric clash with F13 of this peptide, but would not do so for apelin, which features a less bulky proline residue at this position.

**Fig. 2 | Apelin receptor variants show significant differences in expression and peptide ligand binding. a** Experimental pipeline schematic for in vitro characterisation of apelin receptor variants. CHO-K1 cells were transiently transfected with wild-type (WT) or variant apelin receptor constructs tagged C-terminally with an eGFP reporter. In high content screening studies (upper track), cells were plated and treated with fluorescently labelled apelin647 (300 nM) or ELA647 (1 μM) for 90 min before fixation and visualisation. In radioligand binding studies (lower track), cells were lysed and the membranes harvested before treatment with a concentration range (up to 1 nM) of [$^{125}$I]-apelin-13. Bound radioactivity was counted to determine receptor density and affinity. Created in BioRender. Davenport, A. (2024) https://BioRender.com/o44f254. **b** Representative confocal fluorescent images (n ≥ 3 independent experiments). Panels show a merge (eGFP reporter in green and 647 nm fluorescent ligand in red) or the 647 nm fluorescent peptide signal in isolation. Non-specific binding (NSB) was determined in WT cells treated with fluorescent peptide in the presence of 10 μM unlabelled [Pyr$^1$]apelin-13. Scale bars 50 μm. **c** Bar chart showing mean ± SD membrane eGFP fluorescence

(expressed as % WT) from cells imaged over 12 regions, pooled from n = 4 independent experiments in triplicate. Untrans=cells untransfected with apelin receptor cDNA. **d** Bar chart showing mean ± SD count (n = 4 independent experiments in triplicate) of eGFP positive cells per region. **e** Grouped bar chart showing mean membrane apelin647 (solid bars) or ELA647 (slashed bars) fluorescence (expressed as %WT, n = 4 independent experiments in triplicate). For all data expressed in bar charts, statistical significance was determined using a one-way ANOVA, with Dunn's correction for multiple comparisons. *$p < 0.05$, **$p < 0.01$, ***$p < 0.001$, ****$p < 0.0001$. **f** Saturation binding curves showing specific binding of [$^{125}$I]-apelin-13 in transfected membrane preparations. Data are mean ± SD, n = 3 independent experiments. NSB was determined in the presence of 10 μM unlabelled [Pyr$^1$]apelin-13. **g** Competition radioligand binding curves showing ELA-11 peptide competing for binding (% specific) with a single 0.1 nM concentration of [$^{125}$I]-apelin-13 at WT or T/M89$^{2.64}$ variant apelin receptor in membrane preparations. Data expressed as mean ± SD, n = 3 independent experiments. Source data are provided as a Source Data file.

**Table 2 | Summary of pharmacological parameters measured for apelin receptor variants**

| Variant | Expression | | Fl. ligand binding | | Radioligand binding | | | Function | |
|---|---|---|---|---|---|---|---|---|---|
| | Mem. eGFP (% WT) | Object count/ region | Mem. apelin647 (% WT) | Mem. ELA647 (% WT) | $K_D$ (nM) | $B_{max}$ (fmol/ mg) | Hill slope (nH) | pD$_2$ | $E_{max}$ (ΔPWV) |
| WT | 100.00 ± 23.52 | 17.00 ± 6.88 | 100.0 ± 27.09 | 100.00 ± 19.10 | 0.21 ± 0.05 | 236.80 ± 27.07 | 1.26 ± 0.22 | 8.32 ± 0.24 | 39.48 ± 3.93 |
| V/L38$^{1.42}$ | 121.40 ± 31.87 | 16.58 ± 7.99 | 94.74 ± 17.52 | 92.53 ± 28.43 | NBD | NBD | NBD | 7.67 ± 0.48 | 42.55 ± 15.16 |
| G/X45$^{1.49}$ | 32.18 ± 16.67 | 7.67 ± 3.68 | 12.18 ± 6.48 | 20.73 ± 12.55 | NBD | NBD | NBD | – | – |
| T/M89$^{2.64}$ | 104.94 ± 34.40 | 19.83 ± 8.67 | 82.95 ± 24.46 | 31.75 ± 13.20 | 0.25 ± 0.04 | 440.47 ± 38.36 | 1.55 ± 0.22 | 7.72 ± 0.21 | 50.06 ± 8.32 |
| R/H168$^{4.64}$ | 119.93 ± 40.29 | 23.00 ± 10.04 | 2.53 ± 2.10 | 17.29 ± 9.42 | NBD | NBD | NBD | n/a | 5.49 ± 6.42 |
| T/X227$^{5.64}$ | 12.38 ± 3.90 | 5.08 ± 1.51 | 2.94 ± 2.78 | 14.83 ± 7.02 | NBD | NBD | NBD | – | – |
| Untrans. | 0.00 | 0.00 | 0.00 | 0.00 | NBD | NBD | NBD | n/a | 0.21 ± 7.07 |

Colours indicate the missense (V/L38$^{1.42}$, blue; T/M89$^{2.64}$, red; R/H168$^{4.64}$, magenta) or frameshift (G/X45$^{1.49}$, T/X227$^{5.64}$, both yellow) variants.
*Mem.* membrane, *Fl. ligand* fluorescent ligand, *ΔPWV* change in peak wavelength value, *WT* cells transfected with wild-type apelin receptor, *Untrans.* untransfected cells.

Therefore, our modelling complements previous experimental structures to predict the detrimental impact of the variants on endogenous peptide binding, in a manner that is consistent with our pharmacological data showing the reduced binding of ELA at the T/M89$^{2.64}$ variant, and loss of binding of both peptides at the R/H168$^{4.64}$ variant (Fig. 2).

## CMF-019 targets lipophilic sub-pockets at TM interfaces in the apelin receptor orthosteric pocket

To further understand the potential implications of the orthosteric variants for small molecule agonist binding, we determined the crystal structure of the apelin receptor in complex with CMF-019, (Figs. 4, and S4). Importantly, this was enabled by Nxera Pharma UK (Stabilised Receptor) technology (see Methods). Briefly, the apelin receptor NxStaR reported here was generated by introducing six single mutations chosen from a pool produced by either scanning mutagenesis[41,42] or directed evolution[43,44], and further screened using a radioligand binding assay with the small synthetic agonist NXE'870 (compound 48 in Sanofi-Aventis patent #WO2014044738A1)[45]. Only this apelin receptor NxStaR, fused to a bRIL fusion in ICL3, produced sizeable crystallisation hits in Lipidic Cubic Phase (LCP) and in the presence of NXE'870. Further crystallisation condition optimisation, in the presence of CMF-019, produced crystals diffracting beyond 3.0 Å at the microfocus beamline I24 at Diamond Light Source (Harwell, UK). The structure was determined by molecular replacement using the previous apelin receptor crystal structure in complex with AMG3054[31] as a search model and refined against a dataset including reflections up to

~2.6 Å resolution. Residual electron density maps revealed the pose of CMF-019 in the orthosteric pocket, and that of an oleic acid molecule wrapped around TM4 and protruding into the orthosteric pocket via the large opening between TM3, TM4, and TM5.

Our structure showed that the elongated CMF-019 molecule binds in a semi-vertical pose extending from the upper part of the large orthosteric pocket, where it lies at the interface between TM6, TM7, TM4, and ECL2, reaching the bottom of the orthosteric pocket at the interface between TM6, TM7, TM2, and TM3 (Figs. 4, S4, and S5). As a relatively hydrophobic compound (Chrom logD 3.14), CMF-019 makes mostly Van der Waals contacts with the receptor. The isobutyl, thiophene, and isopentyl moieties bind to various hydrophobic side chains lining lipophilic sub-pockets at the interfaces between TM6 and TM7, and TM2 and TM3 (Fig. 4b).

The binding pocket is also lined by several polar residues from TM3, TM4, ECL2, and TM5, notably including the tyrosine residues Y185$^{ECL2}$ and Y264$^{6.51}$ that form hydrogen-bonds with the CMF-019 carboxylic group in the extracellular vestibule (ECV) and core amine, respectively. Interestingly, a significant portion of the orthosteric site, near the interface between TM3, TM4, and TM5, is occupied by an oleic acid molecule. Whilst it corresponds to the monoolein molecule observed in the previously reported AMG3054 and cmpd644 apelin receptor crystal structures, in our structure it protrudes somewhat deeper towards the agonist, and lines the CMF-019 binding site between sub-pocket II and the ECV. Its carboxylic head group stacks onto the CMF-019 aromatic core and forms a salt bridge with R168$^{4.64}$,

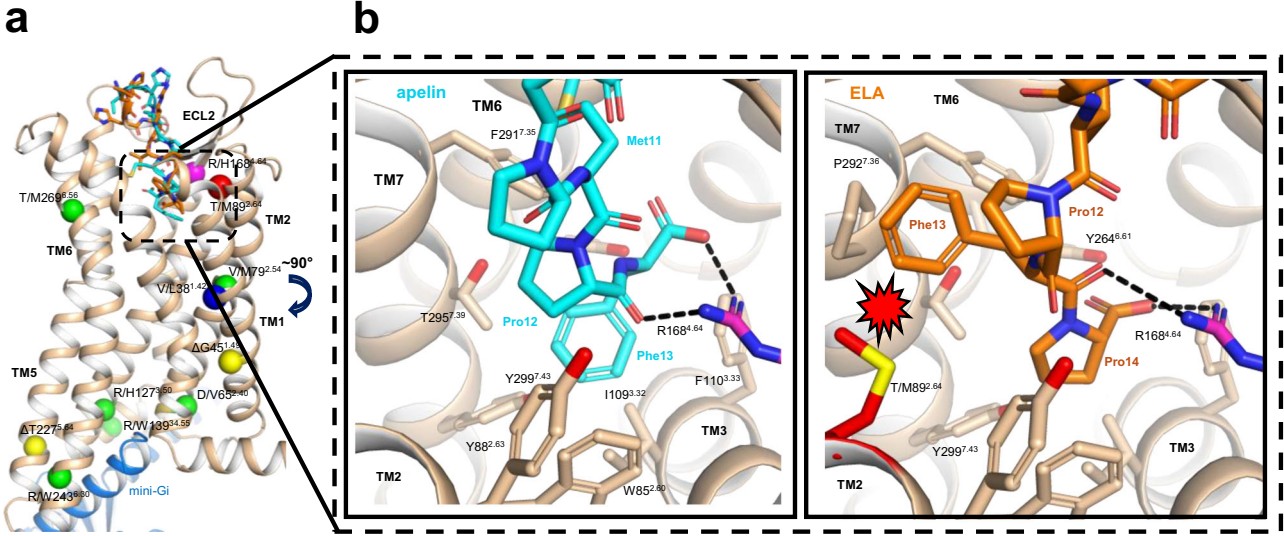

**Fig. 3 | AlphaFold2 models of apelin and ELA bound apelin receptor provide a structural rationale for altered binding at the T/M892.64 and R/H1684.64 variants. a** Cartoon representation of the overall model of human apelin receptor (wheat) bound to mini-Gi (blue). Amino acid positions for variants identified in individuals recruited to the 100,000 Genomes Project are represented as spheres: V38[1.42] (blue), T89[2.64] (red), R168[4.64] (magenta), G45[1.49] and T227[5.64] (yellow), and six others not experimentally characterised (green). The C-termini of apelin (cyan) and ELA (orange) peptides are shown as sticks. **b** Close-up views of apelin (left, cyan) and ELA (right, orange) peptide show C-termini binding in the orthosteric site of the human apelin receptor. Side chains of residues lining the peptide binding pocket, notably R168[4.64] interacting with the C-terminal carboxyl group of the peptides, and the modelled T/M89[2.64] substitution impacting binding of ELA (steric clash indicated by red polygon) but not that of apelin, are represented as sticks. Source data are provided as a Source Data File for 3b.

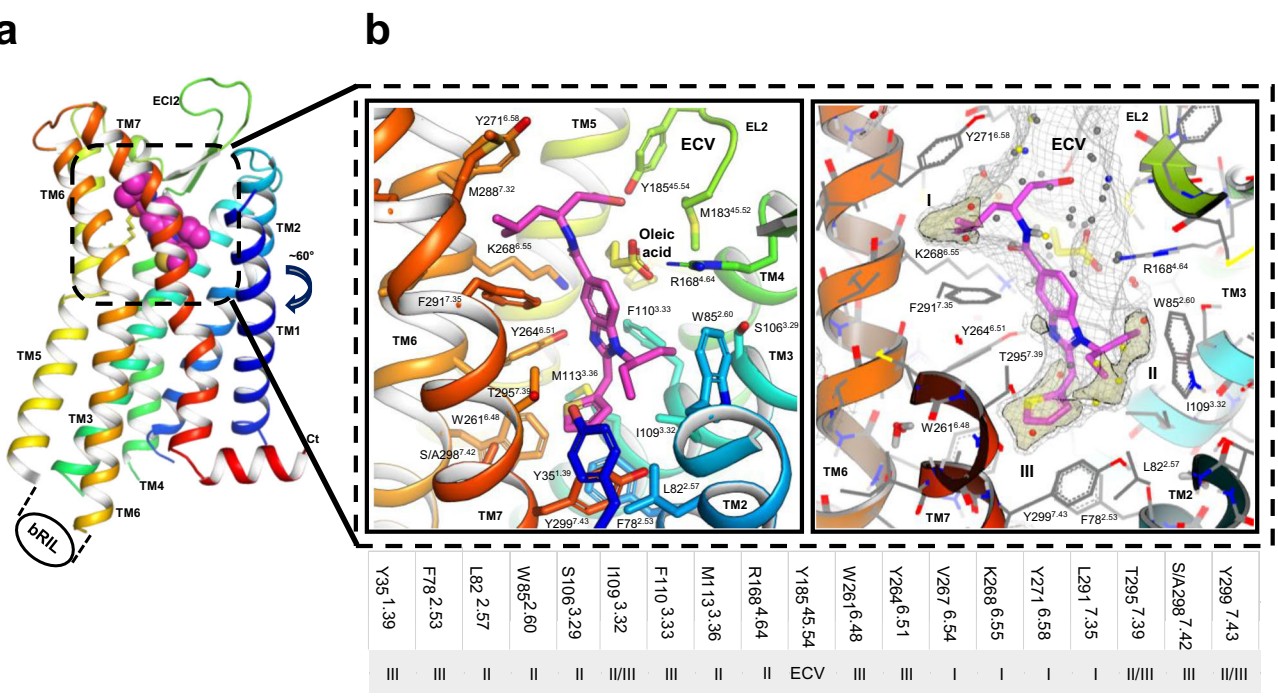

**Fig. 4 | The crystal structure of the apelin receptor NxStaR in complex with CMF-019 reveals deep engagement with lipophilic sub-pockets at the bottom of the orthosteric site. a** Cartoon representation (rainbow colours) of human apelin stabilised receptor (NxStaR) crystal structure (ICL3 bRIL fusion was omitted for clarity), with the CMF-019 ligand represented as spheres, and oleic acid as sticks. **b** Close-up view of the CMF-019 binding pose at the bottom of the apelin receptor orthosteric site, with lining amino acids and oleic acid represented as sticks (left), and CMF-019 binding pocket represented as a light grey mesh, with lipophilic hotspots coloured in pale wheat (right). The extracellular vestibule is labelled as ECV, and the lipophilic sub-pockets are labelled as I, II and III. The small table below panel **b** lists the residues interacting with CMF-019 and the corresponding sub-pockets they line.

which makes a direct contact with an outer edge of the CMF-019 central aromatic ring.

The pose of CMF-019 in our crystal structure is in agreement with that observed in the recent G protein complex cryoEM structure, although there were some notable differences. In the cryoEM structure, CMF-019 is tilted more towards TM1, and its carboxylic moiety does not interact with Y185[ECL2] as observed in the crystal structure, but points instead towards Y271[6.58] (Fig. S5).

Overall, our crystal structure further emphasises how this small synthetic agonist binding mode, reaching towards the bottom of the

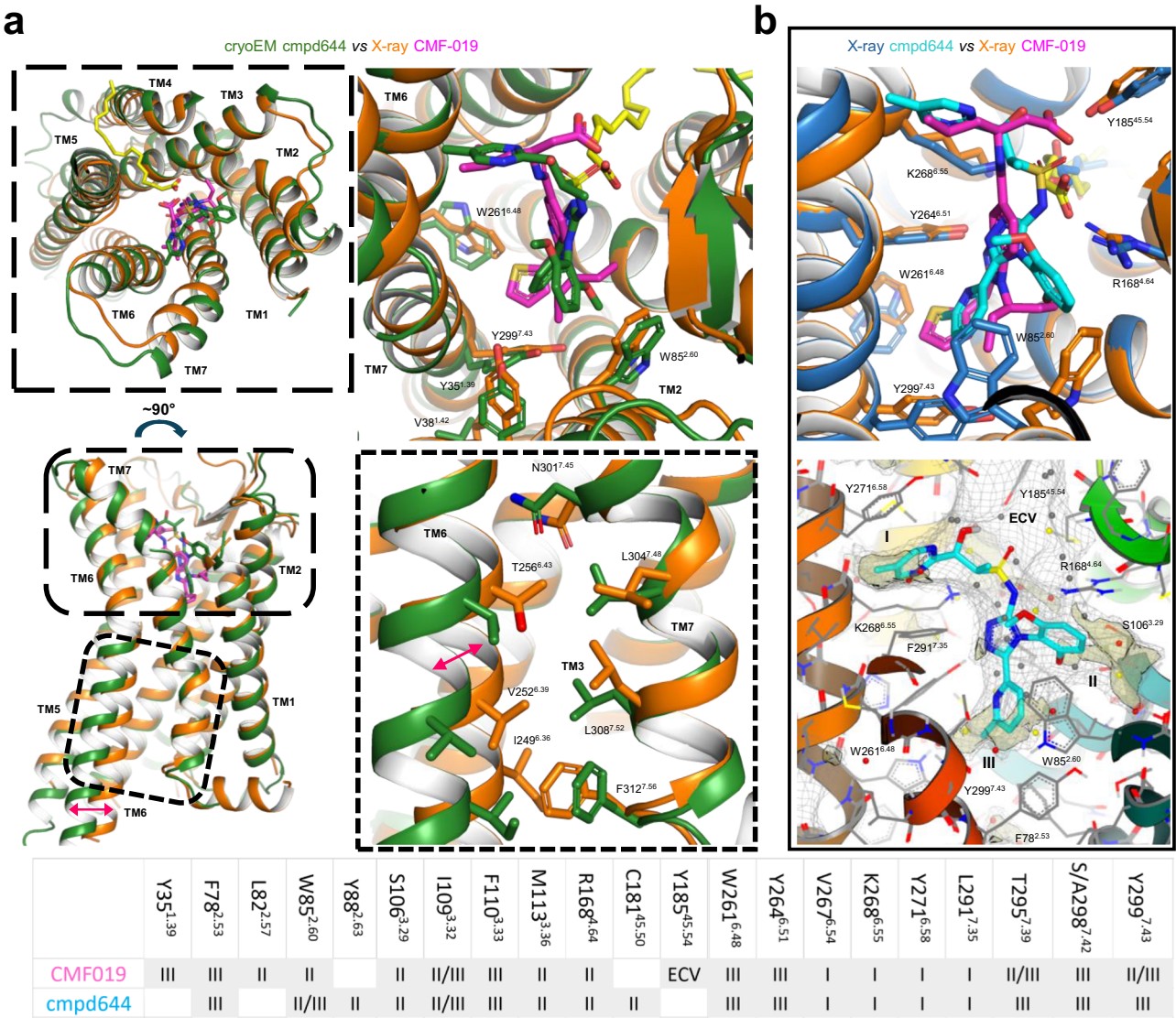

**Fig. 5 | Comparison of small molecule agonist apelin receptor complex crystal and cryoEM structures reveal that the CMF-019 complex exists in an intermediate active state. a** Views of an overlay of the apelin receptor NxStaR crystal structure (orange) in complex with CMF-019 (magenta), with the cmpd644 crystal structure and Gi bound (not shown) cryoEM structure (green) in cartoon representation. Small molecule agonists and relevant amino acid side chains are represented as sticks. Top panels show top-down views, lower left panel shows a side view, and lower right panel shows a close-up view on the TM6-TM7 interface. **b** (Upper) Close-up view of an overlay of the small molecule agonist bound crystal structures, in cartoon representation, with CMF-019 (dark purple), cmpd644 (blue), and relevant amino acid side chains represented as sticks. (Lower) Close-up view of the cmpd644 binding pose in the apelin receptor orthosteric site, with lining amino acid side chains represented as sticks and the cmpd644 binding pocket represented as a light grey mesh with lipophilic hotspots coloured in pale wheat. The extracellular vestibule is labelled as ECV and lipophilic sub-pockets are labelled as I, II and III. The small table lists the residues interacting with CMF-019 or cmpd644, and their corresponding sub-pockets.

orthosteric site and targeting lipophilic sub-pockets at TM interfaces, differs significantly from that of peptides observed in previous apelin and ELA bound experimental structures and in our AlphaFold2-generated models. A two-dimensional representation of the interactions between CMF-019 and the apelin receptor NxStaR are provided in the Supplementary Information (Fig. S6).

### The apelin receptor NxStaR CMF-019 complex shows a conformation reminiscent of an active state

Next, we compared our apelin receptor NxStaR CMF-019 complex crystal structure with the previous cmpd644-bound inactive crystal and G protein-bound active cryoEM structures[29], as well as the recent CMF-019 cryoEM structure[33].

Overall, our crystal structure showed an intermediate conformation that is reminiscent of an active state in the absence of full

activation by G protein binding (Fig. 5, S5, and S7), as observed, for example, in the adenosine $A_{2A}$ receptor crystal bound to the full agonist, NECA[46]. This was remarkably similar to the G protein-bound structures around the orthosteric pocket, but, as expected, some differences were observed in the relative orientation of the intracellular ends of TM5, TM6, TM7, and TM1. Specifically, our apelin receptor NxStaR structure featured a slightly less outward orientation of the cytoplasmic end of TM6 when compared to the G-protein-bound active cryoEM structures, and differences at the interface between TM6 and TM7 (e.g. I249[6.36], V252[6.39], T256[6.43], L304[7.48], L308[7.52], F312[7.56]) for which conformational differences are typically associated with differences in receptor states (Fig. 5b, lower panel).

On the extracellular side, the differences seem to relate to cmpd644 featuring bulkier moieties at the interfaces between TM6 and TM7, and TM7 and TM1. Interestingly, despite CMF-019 and

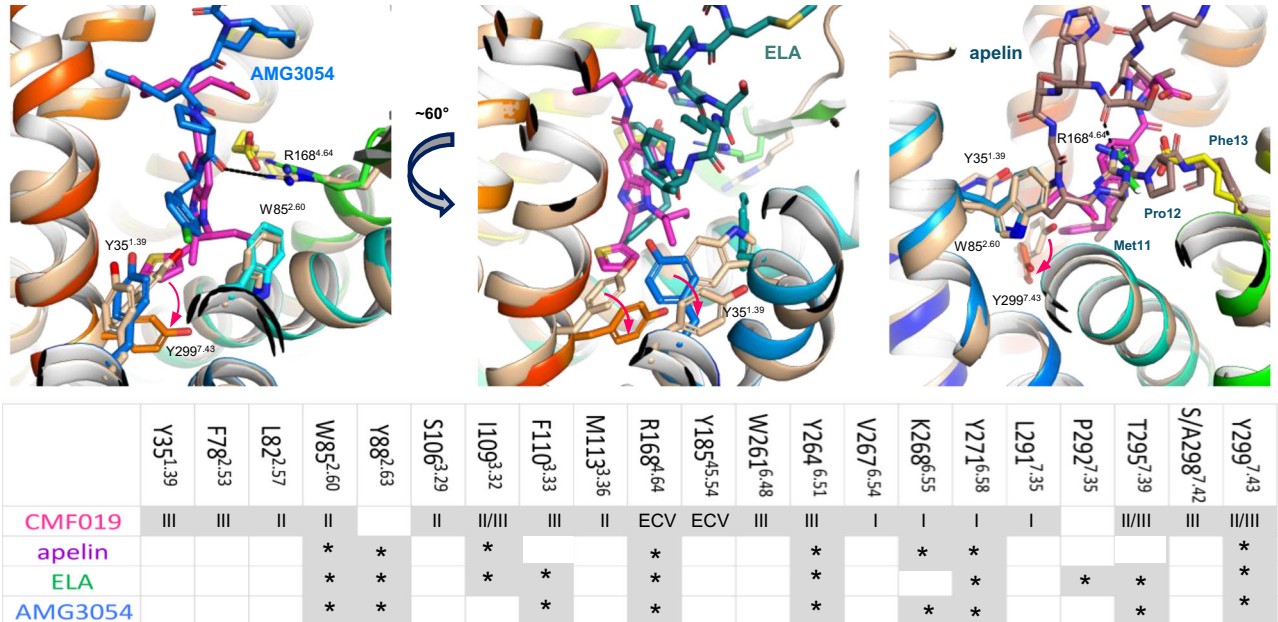

| | Y35$^{1.39}$ | F78$^{2.53}$ | L82$^{2.57}$ | W85$^{2.60}$ | Y88$^{2.63}$ | S106$^{3.29}$ | I109$^{3.32}$ | F110$^{3.33}$ | M113$^{3.36}$ | R168$^{4.64}$ | Y185$^{45.54}$ | W261$^{6.48}$ | Y264$^{6.51}$ | V267$^{6.54}$ | K268$^{6.55}$ | Y271$^{6.58}$ | L291$^{7.35}$ | P292$^{7.35}$ | T295$^{7.39}$ | S/A298$^{7.42}$ | Y299$^{7.43}$ |
|---|---|---|---|---|---|---|---|---|---|---|---|---|---|---|---|---|---|---|---|---|---|
| CMF019 | III | III | II | II | | | II | II/III | III | ECV | ECV | III | III | | I | I | I | I | II/III | III | II/III |
| apelin | | | | * | * | | * | | | * | | | * | | * | * | | | | | * |
| ELA | | | | * | * | | * | * | | * | | | * | | | * | | | * | * | * |
| AMG3054 | | | | * | * | | | * | | * | | | | | * | * | | | | * | * |

**Fig. 6 | Comparison of the CMF-019 complex apelin receptor structure with peptide bound complexes emphasises the deeper reach of the small molecule agonist at the TM6-7 and TM7-2 interfaces.** Panels show overlay of the apelin receptor NxStaR crystal structure (rainbow colours) in complex with CMF-019 (magenta) with the previously reported AMG3054 (blue) bound crystal structure (left), the higher resolution cryoEM ELA (dark teal) G protein bound monomeric complex (7W0P, middle), and the apelin (brown) bound apelin receptor complex (right). Aromatic cage amino acids are represented as sticks. The small table lists the residues interacting with CMF-019, and corresponding sub-pockets as shown in Fig. 3, apelin, ELA, and AMG3054.

cmpd644 being of different chemotypes, they show overlapping binding sites and very similar binding modes, with hydrophobic moieties binding to the lipophilic sub-pockets I, II, and III (Fig. 5b, lower panel), as described above.

The major difference, however, is how deep these agonists bind in the different sub-pockets at the bottom of the orthosteric site. The bulkier CMF-019 thiophene moiety binds deeper into sub-pocket III at the TM6-TM7 interface. However, cmpd644 presents more bulkiness in sub-pocket I, higher at the interface between TM6 and TM7, and its dimethoxy-phenyl moiety binds much deeper into sub-pocket II at the TM2-TM3 interface. It is within this area of the orthosteric site that notable differences for side chains are observed due to some induced-fit. W261$^{6.48}$ and Y299$^{7.43}$ in sub-pocket III show different rotamers, due to the differences in moiety chemical structure between these small molecule agonists. Also, the W85$^{2.60}$ side chain shows two very distinct conformations, one facing the TM1-TM2 interface in the inactive crystal structure, and the other lining the equivalent of sub-pocket II at the TM2-TM3 interface in the active G protein-bound cryoEM structure, which is also the W85$^{2.60}$ side chain conformation observed in our apelin receptor NxStaR CMF-019 structure.

Another major difference was observed around R168$^{4.64}$ at the bottom of the ECV. This residue makes a cation-pi interaction with the cmpd644 dimethoxy-phenyl moiety in the inactive crystal structure, but this interaction is not observed in the active G protein-bound state, because the dimethoxy-phenyl has moved more towards the TM1-TM2 interface. The most striking similarity is observed between sub-pocket II and the ECV, despite the difference in core structure between these small molecules. Whilst the cmpd644 core is built in part around a sulfonamide, this acid group ends up in a position similar to that of the CMF-019 carboxylic group and the head group of the oleic acid molecule.

### Small molecule agonists bind deeper at TM interfaces versus peptides

We further compared our apelin receptor NxStaR CMF-019 crystal structure with previously reported apelin receptor-peptide bound experimental structures (Fig. 6). Overall, peptides engaged with the receptor via the large space at the interface between TM7, TM2, TM3, and ECL2, with their C-termini reaching down in the orthosteric pocket at a site lying at the interface between TM3, TM5, TM6, and TM7 (Fig. S2). This part of the orthosteric pocket is lined, for instance, by the side chains of residues F110$^{3.33}$, Y264$^{6.51}$, and K268$^{6.55}$ (Fig. S2). Compared to the small molecule agonist complex structures, this corresponds roughly to where the small agonist cores are positioned. In this configuration, the conformation of the ELA F10 side chain across the cryoEM structures, for example, is such that it visited either sub-pocket II or III, and the location of the C-terminal carboxylate overlapped with that of the oleic acid head group. Similar to the recent apelin bound cryoEM structure, the M11 side chain overlapped with the CMF-019 thiophene moiety, and P12 and F13 locations overlapped with that of the oleic acid.

However, striking differences were observed for the small molecule versus peptide binding sites. Notably, the small molecule agonists appeared to reach deeper towards the bottom of the orthosteric site, especially at the TM6-TM7 interface. Across these structures, the most noteworthy side chain rotamer differences observed at the bottom of the orthosteric site were for residues Y35$^{1.39}$, F78$^{2.53}$, W85$^{2.60}$, F110$^{3.33}$, and Y299$^{7.43}$, and the small molecule agonists in active conformational states stabilised a rotamer of Y299$^{7.43}$ that is lodged into the interface between TM7 and TM2 (sub-pocket III). Additionally, as mentioned above, the R168$^{4.64}$ side chain showed different positions and conformations, with some positioning the guanidinium group of R168$^{4.64}$ closer to peptide C-termini, so that it interacted with backbone oxygens. In the recent apelin cryoEM structure, such an interaction is only observed with the peptide S6 sidechain. Of interest, the R168$^{4.64}$ residue only formed a minor direct interaction with the core of CMF-019, engaging mostly indirectly via a salt bridge with the oleic acid molecule that was observed to protrude into the binding site (Fig. 4).

In summary, our comparisons showed that CMF-019 binding sites overlap in part with that of the C-terminal residues of the endogenous peptides, but the small molecules reach deeper down the orthosteric

pocket to bind into lipophilic sub-pockets at the interfaces between TM6, TM7, TM1, TM2, and TM3. Two-dimensional plots of CMF-019 and apelin peptide interactions are provided in the Supplemtary Information (Figs. S6 and S8, respectively). Additionally, a summary table showing apelin receptor residues that interact with the small synthetic agonists CMF-019 and cmpd644, and the endogenous peptides apelin and ELA is provided in the Supplemtary Information (Table S1).

## R168$^{4.64}$ tightly regulates binding of peptides but not small molecule agonists

We used surface plasmon resonance (SPR) to measure the binding affinity of apelin-13 and a series of C-terminally truncated isoforms at our apelin receptor NxStaR in detergent (Fig. 7a). Robust binding was observed for apelin-13 and retained for the ΔC2 truncated peptide (ending with M11). However, binding was completely lost for even shorter peptides. This is in agreement with previous findings assessing the potencies of truncated apelin peptides in functional assays, which found that truncation beyond M11 caused loss of function at the apelin receptor[47], and our data show therefore that the apelin receptor NxStaR retains properties of the wild-type protein. Additionally, the data provide further evidence for the importance of interactions at the bottom of the orthosteric site where the receptor binds the last three peptide residues, which is also the binding site for small synthetic agonists.

Next, SPR was used to measure the binding affinities of peptides and small synthetic agonists to the apelin receptor NxStaR versus the R/H168$^{4.64}$ variant (Figs. 7b, S9a). Apelin-13 and the small synthetic agonists, CMF-019 and NXE'870, bind to the wild-type apelin receptor with affinities in the sub-nanomolar to low nanomolar range. However, we could not detect specific binding with ELA as this peptide was found to be too sticky for use in this assay and instead tested NXE'992, a chimeric peptide that combines the five N-terminal residues of [Pyr$^1$] apelin-13 peptide and the four C-terminal residues of ELA, linked by two unnatural amino acids and a serine (pyrazole-QRPRL[AIB] S[PipALA]VPFP) (Fig. S9b).

At the R/H168$^{4.64}$ variant, low nanomolar binding affinities for the apelin peptide and small molecule agonists were retained, but total loss of binding for the chimeric peptide was observed. To test the potential role of the positive charge at the R/H168$^{4.64}$ variant position, the same experiments were performed at pH 6.0, to favour the protonated state of the histidine residue. Whilst binding affinities for the small molecule agonists were consistently higher, binding of the peptides was completely lost. A summary table of SPR data is provided in the Supplementary Information (Table S2).

These findings confirm that the R168$^{4.64}$ residue is an important regulator of peptide binding at the apelin receptor but has very little impact on the binding of the small molecule agonists, CMF-019 and NXE'870, when substituted with a histidine residue, as occurs in the naturally occurring R/H168$^{4.64}$ human variant.

## The R/H168$^{4.64}$ apelin receptor variant induces a detrimental phenotype in hESC-CMs

Following the findings that the R/H168$^{4.64}$ apelin receptor variant caused loss of binding and function in vitro (Fig. 2, Table 2), and that R168$^{4.64}$ regulated endogenous peptide engagement in structural and SPR studies (Figs. 3, 6, and 7), we used CRISPR base editing to introduce the variant into a human embryonic stem cell derived cardiomyocyte (hESC-CM) model (Fig. 8a). Expression analysis confirmed both the mRNA and protein levels of the wild-type and R/H168$^{4.64}$ variant were similar in both undifferentiated stem cells and differentiated hESC-CMs following the base editing treatment (Fig. 8b). Consistent with previous data, binding of both apelin647 fluorescent ligand and [$^{125}$I]-apelin-13 radioligand was significantly reduced in the

R/H168$^{4.64}$ line versus wild-type background in both stem cells and hESC-CMs (Fig. 8c, d).

Despite consistent receptor expression, the loss of apelin binding at the R/H168$^{4.64}$ variant highlighted the potential to induce a phenotypic change that was explored in several assays (Fig. 8e). Firstly, the overall differentiation efficiency of the cells was significantly reduced, with 80 ± 9% of wild-type cells staining positive for the cardiac-specific marker troponin T, versus 52 ± 6% in the R/H168$^{4.64}$ variant line. Staining for the fibroblast-specific marker, Thy1, did not significantly differ between the two conditions (Fig. S10), indicating that the R/H168$^{4.64}$ cells were not being driven towards this lineage.

Secondly, apelin peptide secretion into culture medium was detected by ELISA, and showed that the R/H168$^{4.64}$ line secreted significantly more peptide (0.30 ± 0.10 ng/mL) versus wild-type (0.10 ± 0.04 ng/mL). This may reflect a compensatory mechanism for the dysfunctional receptor, or might indicate that the R/H168$^{4.64}$ cells are retaining a more stem-like phenotype that shows higher levels of secretion than would be expected for differentiated cardiomyocytes.

Finally, voltage sensing studies showed that the R/H168$^{4.64}$ hESC-CMs exhibited a prolonged time-to-peak voltage (206.57 ± 14.88 ms) versus wild-type (74.42 ± 23.72 ms). While the data point to a direct effect of the variant on electrophysiology in the heart, the prolonged time-to-peak voltage may again be a result of the reduced cardiac differentiation efficiency. Regardless of the mechanism, the R/H168$^{4.64}$ hESC-CMs evidently perform more poorly than wild-type cells, and the data reinforce the critical role that the apelin receptor plays in heart formation and function.

## Discussion

Genetic variation has been shown to occur in GPCRs, including at endogenous peptide and synthetic molecule binding sites, and at interfaces with signalling proteins[35]. Therefore, GPCR coding variants are an important factor in the differentiation of individual responses to endogenous ligands, potentially leading to pathophysiological conditions and to possible effects on the efficacy of therapeutic drugs[35]. The research presented here focussed on naturally occurring coding variants of the apelin receptor identified in the UK Genomics England 100,000 Genomes Project. Our experimental approach to characterise these variants combined in vitro pharmacology, molecular modelling, SPR, and gene editing in stem cell-derived cardiomyocytes. Additionally, we leveraged Nxera Pharma NxStaR® technology to enable the crystal structure determination of the apelin receptor in complex for the first time with a drug-like G protein-biased agonist, CMF-019. Overall, we gained insights into key determinants of endogenous and drug-like agonist engagement at this important receptor with broad therapeutic potential.

Several of the variants identified exhibited altered pharmacologies compared to the wild-type receptor. Overall, the frameshift mutations G/X45$^{1.49}$ and T/X227$^{5.64}$ showed significantly altered expression and localisation at the cell surface compared to wild-type, but receptors produced were inactive in peptide binding (Fig. 2, Table 2). In contrast, the missense variants V/L38$^{1.42}$, T/M89$^{2.64}$, and R/H168$^{4.64}$ were expressed and localised like the wild-type, but showed differentiated binding responses to peptides and small molecules, highlighting possible key determinants of ligand engagement.

The previously reported structures of the apelin receptor, notably bound to ELA and apelin, provided some insights into the detrimental effects of these variants on peptide engagement. ELA binds on the extracellular side of the large orthosteric pocket via the interface between TM7, TM1 and TM2, where T89$^{2.64}$ is found, with the ELA C-terminus reaching towards the interface between TM3, TM5, and TM6, and binding to a site lined notably by R168$^{4.64}$[29,31]. However, because the positions of T89$^{2.64}$ or R168$^{4.64}$ relative to the peptides are not consistent across all previous experimental structures, and a

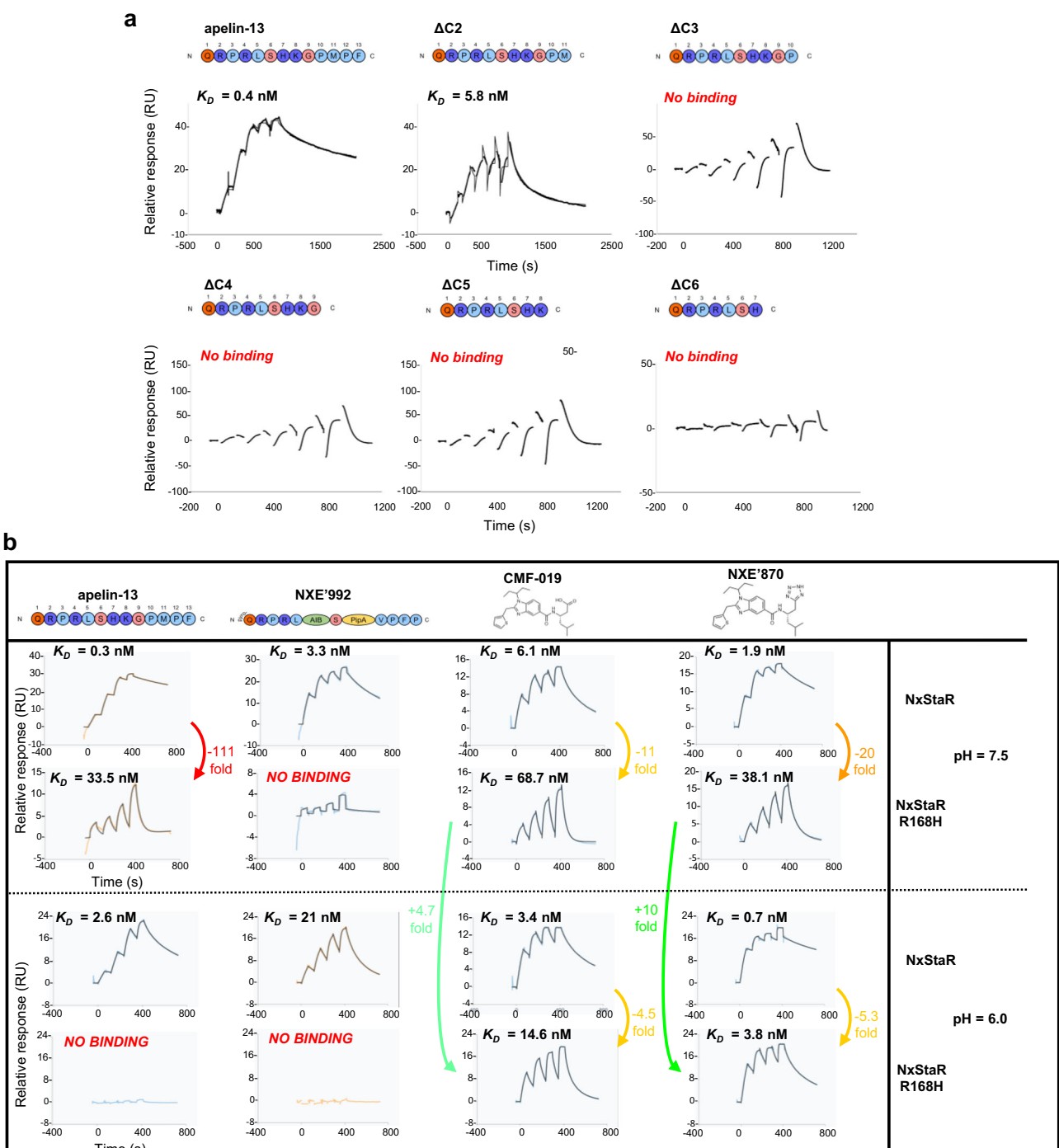

**Fig. 7 | Surface plasmon resonance (SPR) sensorgrams of the apelin receptor NxStaR show loss of binding of truncated peptides and that the R/H168$^{4.64}$ variant can still bind small molecule agonists. a** SPR sensorgrams of the apelin receptor NxStaR in the presence of endogenous apelin-13 peptide versus C-terminally truncated peptides (ΔC2-ΔC6). **b** SPR sensorgrams of apelin receptor NxStaR or the R/H168$^{4.64}$ variant NxStaR in the presence of endogenous apelin-13 peptide (first column), the chimeric peptide NXE'992 (second column), the G protein biased, small molecule agonist CMF-019 (third column), and NXE'870 (fourth column) at pH 7.5 or 6.0. Source data are provided as a Source Data file.

structure with apelin was not available when our experimental work was carried out, we used AlphaFold2 as a complementary approach to model complexes with ELA and apelin, as well as the T/M89$^{2.64}$ variant, providing a more complete picture of endogenous peptide binding that explains the detrimental effects of the variants. Our models reveal further that the T/M89$^{2.64}$ variant would greatly narrow the interface between TM7 and TM2, creating a steric clash with ELA, particularly at F13 (Fig. 3), but less so with apelin, which presents a less bulky proline

at this interface, explaining the differentiated effect of the T/M89$^{2.64}$ variant on ELA and apelin binding.

Additionally, our models show agreement with the previous peptide structures (Figs. 3, and S2) that R168$^{4.64}$ can interact extensively with the peptide backbone. In fact, R168$^{4.64}$ is known to be key for endogenous peptide binding, as the artificial R/A168$^{4.64}$ substitution has been shown to abolish peptide binding[29,31]. This may be common across other GPCRs, where, for example, an artificial

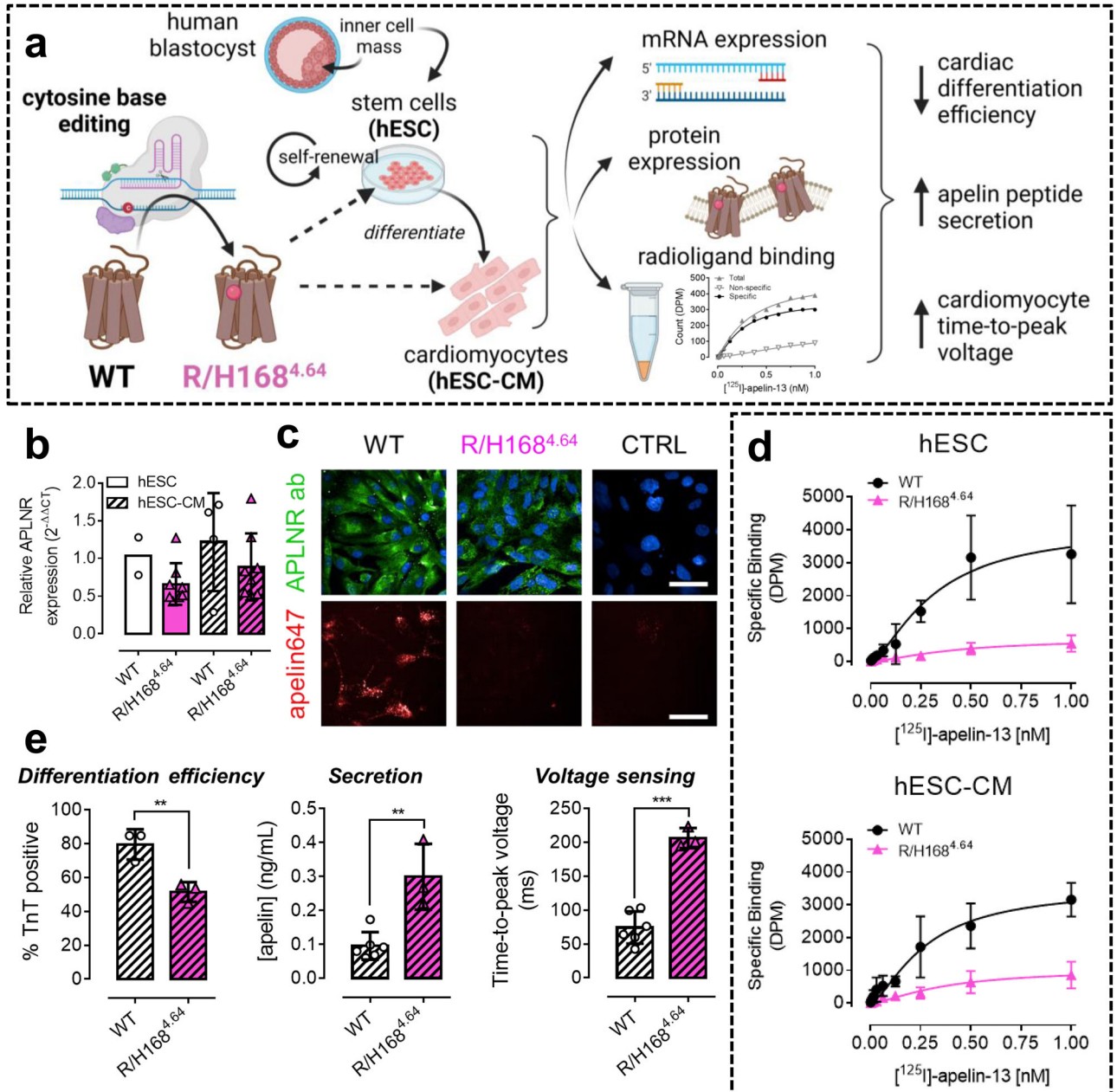

**Fig. 8 | Using base editing to introduce the R/H168[4.64] substitution into endogenously expressed apelin receptor reduces binding and induces phenotypic dysfunction in a human hESC-CM model. a** Schematic showing experimental pipeline. Cytosine base editing technology was used to introduce the R/H1684.64 substitution into endogenously expressed apelin receptor in human embryonic stem cells (hESCs) originally harvested from the inner cell mass of a human blastocyst, or into differentiated cardiomyocytes (hESC-CMs). Cells were characterised for R/H168[4.64] apelin receptor mRNA expression, protein expression, and radioligand binding. Functional consequences of the R/H1684.64 substitution were also assessed. Created in BioRender. Davenport, A. (2024) https://BioRender.com/e87y890. **b** Relative apelin receptor gene (APLNR) expression in wild-type (WT) or R/H168[4.64] variant hESC (solid bars) and hESC-CM (slashed bars) lines. Data expressed as $n = 2$ for WT hESC, $n = 4$ (mean ± SD) for WT hESC-CMs, and $n = 8$ (mean ± SD), for R/H168[4.64] hESC and hESC-CMs. **c** Representative confocal fluorescent images ($n = 4$ independent experiments) of WT and R/H1684.64 hESC-CMs treated with anti-apelin receptor antibody (visualised in green) and nuclear marker

(visualised in blue) (upper panels) or 300 nM apelin647 fluorescent ligand (visualised in red) (lower panels). Control WT hESC-CMs (upper) were treated with secondary antibody in the absence of the primary, or (lower) in the presence of 10 μM unlabelled [Pyr[1]]apelin-13. Scale bars indicate 50 μm. **d** Saturation radioligand binding curves showing specific binding of [[125]I]-apelin-13 in hESC (upper) or hESC-CM (lower) membrane preparations. Non-specific binding was determined in the presence of 10 μM [Pyr[1]]apelin-13. Data expressed as mean ± SD, $n = 3$ independent experiments. **e** Functional consequences of the R/H1684.64 substitution in hESC-CMs. Bar charts show differentiation efficiency measured by % hESC-CMs staining positive for troponin T (TnT) (left), apelin peptide concentrations secreted from hESC-CMs as determined by ELISA (middle), and time-to-peak (TTP) voltage per unit time (ms) (right). For all panels here, data are expressed as mean ± SD, $n \geq 3$ independent experiments. Statistical significance was determined using a one-way ANOVA, with Tukey's correction for multiple comparisons looking for differences between wild-type (WT) and variant (R/H168[4.64]) apelin receptor. *$p < 0.05$, **$p < 0.01$, ***$p < 0.001$. Source data are provided as a Source Data file.

substitution at this amino acid position (R/C167[4.64]) abolished binding of endogenous peptide at the angiotensin II type 1 (AT$_1$) receptor[48]. Strikingly, we observe here a complete loss of peptide binding, even for a conservative arginine to histidine replacement, further emphasising the importance of hydrogen-bonding interactions formed by the guanidinium group of R168[4.64] with the peptide backbone. Moreover, this binding mode involving the peptide backbone also explains why the loss of binding is not sequence-specific, thus affecting both apelin and ELA at the R/H168[4.64] variant. This AlphaFold2 modelling complements previous experimental structures to predict the detrimental impact of the variants on endogenous peptide binding, consistent with our in vitro data showing notably reduced binding of ELA only at the T/M89[2.64] variant, and of both peptides at the R/H168[4.64] variant.

The crystal structure of the apelin NxStaR in complex with CMF-019 presented here, in agreement with the recent CMF-019 cryoEM structure, expands our understanding of binding modes of small synthetic agonists and emphasizes strongly differentiated binding modes between peptides and drug-like agonists. Despite their different chemotypes, CMF-019 and cmpd644 share overlapping binding sites, as well as binding modes involving mostly hydrophobic contacts. Not unexpectedly, this synthetic agonist binding site overlapped with that of hydrophobic and C-terminal residues of apelin and ELA, suggesting that, within the large orthosteric site of this peptidergic receptor, this location is an important site for agonist peptides, as supported by our SPR data using truncated apelin peptides (Fig. 7a). However, small synthetic agonists likely compensate for their small size compared to peptides by reaching even deeper to bind to lipophilic sub-pockets at interfaces between transmembrane helices, such as TM6 and TM7, for which conformational changes are typically associated with GPCR activation.

Another key difference with peptides is the different side chain conformation of Y299[7.43], which has been linked to bias. Here, we observed CMF-019 stabilising a rotamer of Y299[7.43] (Fig. 5a) that is distinct from the rotamer observed in the peptide complex structures (Fig. 5b). The Y299[7.43] residue is conserved among other class A GPCRs, where it plays an important role as a 'bias switch'. For instance, ligand-induced conformational changes in the corresponding Y291[7.43] residue in the chemokine receptor CCR1 allosterically regulate β-arrestin activation to drive signalling bias[49]. Additionally, mutagenesis in the AT$_1$ receptor has previously demonstrated an impairment of a Y/F292[7.43] mutant to recruit β-arrestin in response to a benchmark β-arrestin-biased ligand[50]. Therefore, binding in this sub-pocket III at the interface between TM6 and TM7 may contribute to G protein-bias in the apelin receptor, and warrants further investigation in order to inform the design of biased drugs that might offer better therapeutic outcomes.

Next, we aimed to pharmacologically characterise the R/H168[4.64] variant in a more relevant setting in human cells by using CRISPR base editing to introduce the variant into a hESC-CM model (Fig. 8). Whilst the R/H168[4.64] variant was expressed similarly to the wild-type receptor, its engagement with ligands was significantly reduced, consistent with our data in CHO-K1 cells (Fig. 2, Table 2). Furthermore, the R/H168[4.64] variant induced a perturbed phenotype in the hESC-CMs. Differentiation efficiency was significantly reduced. However, time-to-peak voltage in cells that were visibly contracting and therefore differentiated, was prolonged compared to wild-type, closely phenocopying a previously reported inducible apelin receptor knockdown system in the same cell model[51]. Intriguingly, in this model[51], the SCN5A, (the gene encoding the pore forming α-subunit of the voltage-dependent cardiac sodium channel NaV1.5) was the most significantly downregulated differentially expressed 'druggable' gene that could be experimentally studied. This is consistent with the hypothesis for a role for apelin receptor signalling in modulating cardiomyocyte sodium currents[11]. The data highlight the potential of using genetic editing to generate 'disease-in-a-dish' models for the apelin receptor.

Finally, one of our key observations was that the R168[4.64] residue is not a key contributor to the binding of CMF-019 (Fig. 4), in agreement with the previous active cryoEM structure with cmpd644, and our SPR data further confirmed that the R/H168[4.64] variant retains nanomolar binding affinity for small synthetic agonists (Fig. 7b). It is interesting to consider that small synthetic ligands might be able to bind and exert agonist activity in individuals harbouring variants that reduce or even abolish endogenous ligand binding, potentially providing a personalised medicine strategy to rescue disease phenotypes in these patients.

GPCRs form the largest class of therapeutic drug target for currently approved medicines. Identifying the mechanisms of ligand engagement at GPCRs will lead to the discovery of new medicines targeting these highly tractable drug targets[2–4] In this study we report a series of apelin peptides. Our data confirm that G protein biased apelin peptides retain the beneficial in vivo cardiovascular actions of the endogenous apelin peptides and we have identified NXE'065 which has greater cAMP versus β-arrestin pathway selectivity than CMF-019 (Figs. S11, and S12, Table S5). Here, we have used a combinatorial approach, encompassing the identification and in vitro characterisation of naturally occurring rare receptor variants, modelling, and a crystal structure bound to a G protein-biased small molecule, to pinpoint key structural determinants of endogenous peptide and synthetic ligand binding and activity at the apelin receptor. The data provide new insights that might facilitate and guide structure-based drug design at this physiologically and pathophysiologically relevant GPCR.

## Methods
### Ethics declaration
Ethical approval and written informed consent was obtained from the participants by the National Institute for Health Research (NIHR) BioResource for Rare Diseases[34]. All animal care and rodent experiments complied with the Home Office (United Kingdom) guidelines under the Animals (Scientific Procedures) Act 1986 Amendment Regulations (SI 2012/3,039) and were approved by the local ethics committee (University of Cambridge Animal Welfare and Ethical Review Body).

### Identifying rare sequence variation in the apelin receptor (APLNR) gene
Whole genomes sequenced by the NIHR BioResource (NBR) project, a prospective component of the Genomics England 100,000 Genomes Project, were screened for rare sequence variants in the apelin receptor (APLNR) gene that had a minor allele frequency of <1 in 10,000 in the general population. The cohort comprised 13,037 patients diagnosed with one of several rare diseases or cancers. Following whole genome sequencing, case-control rare variant analysis was performed as described previously[52].

We used predetermined exclusion criteria to identify nine apelin receptor single nucleotide missense variants that occurred at highly conserved sites, predicted using in silico conservation scores from GERP (Genomic Evolutionary Rate Profiling), phastCons, and phyloP (http://genome.ucsc.edu)[53,54], and that were likely to show pathogenicity using deleteriousness scores from SIFT (Sorting Intolerant From Tolerant)[55] and PolyPhen-2 (Polymorphism Phenotyping v2)[56]. Additionally, these variants occurred at sites predicted to affect receptor binding and/or function when compared to corresponding sites in similar class A GPCRs using GPCRdb (https://gpcrdb.org/)[37]. For example, apelin receptor residue R127[3.50] forms part of the E/DRY motif, which is highly conserved amongst class A GPCRs to regulate conformation and activity, and mutation would likely alter protein function[57]. The exclusion criteria ensured only variants that were ultra-rare in the general population, likely under selective pressure, and that

occur at amino acid sites that are highly evolutionarily conserved were retained. After initial filtering, subsequent analyses of genetic data in the gnomAD (version 4.1.0) and UK BioBank cohorts, comprising a total of 807,602 samples, confirmed that selected variants remained at an ultra-rare level of occurrence across these datasets. Overall, nine missense variants, and two single nucleotide deletion frameshift mutations, were subsequently characterised in vitro.

## Apelin receptor plasmid DNA

Wild-type apelin receptor was PCR cloned into pcDNA3.1 (Invitrogen), using Phusion Hot Start II DNA polymerase (ThermoFisher Scientific), in accordance with the manufacturer's instructions. Apelin receptor variants were made by site-directed mutagenesis, using Phusion Hot Start II DNA polymerase (ThermoFisher Scientific), in accordance with the manufacturer's instructions. Receptor plasmid constructs were tagged C-terminally with an eGFP reporter. Cloned plasmid vectors were transformed in DH5α competent *Escherichia coli*, before isolation using a QIAprep Spin Miniprep Kit (QIAGEN), in accordance with the manufacturer's instructions.

## CHO-K1 cell culture and transfection

CHO-K1 (ECACC 85051005) cells were cultured in 25 mL DMEM/F-12 (1:1) nutrient mix + L-glutamine (ThermoFisher Scientific), supplemented with 10% FBS (ThermoFisher Scientific) in the presence of 0.1 mg/mL Normocin antibiotic formulation (InvivoGen) in T175 cell culture flasks (Corning). Cells were maintained at 37 °C, in humidified air containing 5% $CO_2$. Transfection with wild-type or variant apelin receptor plasmid DNA (1 ng/mL final concentration) was performed using a TransIT-CHO Transfection Kit (MIR 2174, Mirus), in serum-free Opti-MEM I media (11058021, Gibco), as per the manufacturer's instructions.

## High content screening

High content screening was used to characterise apelin receptor variant expression, membrane localisation, and fluorescent ligand binding. CHO-K1 cells transiently transfected with wild-type or variant apelin receptor construct tagged C-terminally with eGFP were plated at ~10,000/well in CellCarrier-96 Ultra Plates (PerkinElmer). On the day of experimentation, cells were washed with HBSS before treatment with well-validated apelin647 (300 nM) or ELA647 (1 µM) fluorescent ligand[38] for 90 min at room temperature, until equilibrium was reached. Non-specific binding was determined in the presence of a saturating 10 µM concentration of unlabelled [Pyr¹]apelin-13. Fluorescent ligand was washed out with HBSS, and cells were fixed with 4% formaldehyde for ~3 min. After another HBSS wash, cells were treated with Hoechst 33342 nuclear marker (10 µg/mL) for 15 min. After a final wash, cells were maintained in 100 µL HBSS.

Cells maintained in the 96-well plates were imaged on an Opera Phenix High Content Screening System (PerkinElmer), a highly sensitive, dual-camera spinning disk confocal fluorescent imaging platform, using a 40x/NA1.1 water immersion objective. The imaging field included a grid of 16 (4 × 4) regions, where each region is ~250 × 250 µm², at a focal depth of 0.5 µm above the bottom of the plate. Three fluorescent channels were used. The first channel (blue) used an excitation wavelength of 405 nm and emission filter of 435–480 nm for the Hoechst 33342 nuclear marker. The second channel (green) used an excitation wavelength of 488 nm and emission filter of 500–550 nm for the eGFP reporter. The third channel (red) used an excitation wavelength of 631 nm for the apelin647 and ELA647 fluorescent ligand. For all channels, excitation laser intensity was set at 50% power, with a 50 ms exposure time. Additionally, each imaging field was imaged using bright-field and digital phase contrast (DPC). The DPC images were used to identify outer boundaries of the cells as a surrogate of the membrane, and a collar extending ±2 µm from this membrane region was established to segment this cellular compartment.

Following acquisition, fluorescence was quantified using in-built Harmony High Content Imaging and Analysis Software (PerkinElmer), as described previously[38]. Only cells that were positive for eGFP (i.e. mean whole cell eGFP fluorescence of >120 arbitrary fluorescence units, AFU) were included in the analysis, where background fluorescence was calculated as ~112 AFU. Mean membrane fluorescence was then quantified for the green channel as an indication of membrane protein expression, and for the red channel as an indication of fluorescent ligand binding.

## Radioligand binding

Radioligand binding studies were performed using [¹²⁵I]-apelin-13 ([Glp⁶⁵,Nle⁷⁵,Tyr⁷⁷][¹²⁵I]-apelin-13, PerkinElmer)[10,30,38]. The radiolabel has a specific activity of 2200 Ci/mmol. Plastic-ware used in radioligand binding experiments was coated with Sigmacote, a siliconising reagent that forms a covalent film to reduce sticking of the radiolabel.

Membrane protein preparations were made by harvesting CHO-K1 cells transiently transfected with wild-type or variant apelin receptor construct into tubes. Tubes were spun at 1000 xg for 10 min at 4 °C to pellet cells, before resuspending and triturating in ice-cold TRIS wash buffer (50 mM TRIS-HCl in deionised water, pH balanced to 7.4 at room temperature) to induce complete hypotonic lysis of cells. Cells were spun a second time at $21,000 \times g$ for 20 min at 4 °C, to obtain a crude membrane fraction. Supernatant was discarded and membrane preparations resuspended in TRIS wash buffer. Cell membrane preparations were stored at −70 °C until use in experiments. Protein concentration of cell membrane preparations was determined using a DC Protein Assay (Bio-Rad), as per the manufacturer's instructions.

For binding experiments, membrane preparations were diluted in binding buffer (50 mM TRIS-HCl and 5 mM $MgCl_2$, balanced to pH 7.4 at room temperature) to give a final protein concentration of 1 mg/mL. For saturation binding experiments, protein was treated with a concentration range (2 pM–1 nM) of [¹²⁵I]-apelin-13, diluted in binding buffer, and incubated for 90 min at room temperature to reach equilibrium. For competition binding experiments, protein was treated with a single 0.1 nM concentration of [¹²⁵I]-apelin-13, against a concentration range (1 pM–1 µM) of ELA-11 peptide. Non-specific binding was determined in the presence of a saturating 10 µM concentration of unlabelled [Pyr¹]apelin-13 in all instances. At the end-point, samples were spun in a centrifuge at $20,000 \times g$ for 10 min at 4 °C to terminate equilibrium. Supernatant was aspirated, and pellets were resuspended and triturated in 500 µL ice-cold wash buffer, over ice. Samples were spun a second time at $20,000 \times g$ for 10 min at 4 °C, before aspiration of the supernatant. Radioactivity in pellets was counted using a Cobra II model 5003 gamma counter (Packard).

Saturation binding data were analysed using the EBDA and LIGAND components of the KELL (Kinetic, EBDA, Ligand, Lowry) software package (Biosoft)[58] to generate receptor affinity ($K_D$) and receptor density ($B_{max}$) values. $B_{max}$ values, in fmol/mg, were calculated using the known $K_D$ value of the radiolabel (0.076 nM), known specific activity of the radiolabel (2200 Ci/mmol), and concentration of protein used (1 mg/mL). Saturation data were presented using the nonlinear one site-specific binding with Hill slope model fitted by GraphPad Prism version 6.07 for Windows (GraphPad Software). Competition binding data were presented using the nonlinear one site Fit $K_i$ [2] model fitted by GraphPad Prism, with constraints for the radiolabel concentration (0.1 nM) and $K_D$ (0.076 nM). The software was also used to calculate the binding affinity ($K_i$ value) of competing ELA-11 ligand using the Cheng-Prusoff equation.

## Dynamic mass redistribution functional assay

Functional responses of CHO-K1 cells transfected with wild-type or variant apelin receptor to [Pyr¹]apelin-13 were assessed using BIND reader technology (SRU Biosystems) dynamic mass redistribution assay. Cells were plated at ~25,000/well in proprietary 96-well

biosensor plates that are covered with a nanostructured optical grate. Cells were subsequently washed with HBSS before acclimatising to room conditions for 20 min. The plate was inserted into the BIND reader and a 5 min baseline read was recorded before treatment with a concentration range (0.1 nM–1 μM final) of [Pyr¹]apelin-13. Maximum change in peak wavelength value (ΔPWV) from baseline after drug addition was used to plot concentration response curves, where data were fitted to four parameter logistic curves in GraphPad Prism. Maximum response ($E_{max}$) and $pD_2$ (−log10 $EC_{50}$) values were calculated for [Pyr¹]apelin-13 at wild-type and variant apelin receptors. Untransfected cells were unresponsive to [Pyr¹]apelin-13.

### AlphaFold2 modelling
The active state of apelin receptor bound to peptide ligands was modelled with AlphaFold2 (https://www.readcube.com/library/5d1671d8-5c39-40fd-8d64-7a7d3a0c899e:ccb52d59-1461-4b7a-a85e-1e21631710a1). AlphaFold2 was run in multimer mode using the sequences of the apelin receptor, mini Gi, and either apelin or ELA. Five structures were generated for each language model and the output 25 models were ranked. All models were examined simultaneously and the best representative model by score was selected for analysis. AlphaFold2 was chosen for modelling these structures as the native apelin peptide bound to active apelin receptor has not as yet been determined. While structures of ELA bound to active apelin receptor do exist, the binding mode is variable and the resolution is low. The AlphaFold2 model of ELA at apelin receptor is consistent with the existing structures, is supported by the relevant SAR, and provides a comparative model to the apelin-apelin receptor predicted model. The mutation, T/M89[2.64], was modelled using Schrodinger's Maestro suite (Schrödinger Release 2022-2: Maestro, Schrödinger, LLC, New York, NY, 2021.). The binding surface was calculated using GRID from Molecular Discovery (https://pubmed.ncbi.nlm.nih.gov/3892003/).

### Generation of human apelin stabilised receptor (NxStaR®)
Full-length human apelin receptor was used as a background to generate a thermostabilised receptor using a combination of scanning mutagenesis[41,42] utilising tritiated [³H]-NXE'870 (RC TRITEC), and directed evolution[43,44] utilising [Cys(AF488)]-[PEG4]-QRPRLSHKGP-[Nle]-P-Y(OBn)-acid (Cambridge Research Biochemicals). For assessment of mutant thermostability, cells transfected with receptor were treated with the [³H]-NXE'870 radiolabel for 2 h at room temperature to reach equilibrium, before termination by incubation at 4 °C for 5 min. The compound chemotype is similar to CMF-019[30]. Cells were then solubilised with 0.1% (w/v) n-dodecyl-β-d-maltopyranoside (DDM) supplemented with 0.12% cholesteryl hemisuccinate and 0.6% (w/v) 3-(3-cholamidopropyl) dimethylammonio-1-propanesulfonate (CHAPS) (Anatrace) for 1 h at 4 °C. Crude lysates were cleared by centrifugation at 16,000 × $g$ for 15 min. Thermostability was measured by incubation of lysate samples at different temperatures for 30 min, followed by separation of unbound radioligand using gel filtration. Levels of ligand bound receptor were determined using liquid scintillation counting. Thermal stability (apparent Tm) was defined as the temperature at which 50% ligand binding was retained following plotting of the radioligand binding data against temperature using the sigmoidal dose-response (variable slope) equation in GraphPad Prism. The resultant apelin receptor NxStaR contained six thermostabilising mutations: T/V87[2.62], N/A112[3.35], T/M207[5.44], F/L214[5.51], I/A224[5.61], and S/A298[7.42]. The NxStaR thermostabilising mutations were functionally assessed, showing comparable expression and binding compared to the wild-type apelin receptor template but overall improved thermostability (see Table S3).

To generate the minimal construct required for crystallisation, the apelin receptor NxStaR was truncated at the N- and C-termini (leaving receptor residues 7-330), and two palmitoylation sites (C/L325 and C/M326) and one N-linked glycosylation site (T/N177) were mutated. To identify the most optimal bRIL (E. coli b₅₆₂RIL domain) fusion construct, a matrix was designed between I228 and E238 in ICL3, with 30 constructs generated. These constructs were ranked using thermostability and fSEC analysis[59], and the most optimal fusion construct was found to comprise a bRIL between I228 and E235.

### Apelin receptor NxStaR expression, membrane preparation, and protein purification
Towards protein preparation, we found a combination of single mutations and truncations that improves expression of the apelin receptor in insect cells, and its thermostability in the presence of the small synthetic agonist NXE'870 (Fig. S7). The compound chemotype is similar to CMF-019[30]. For crystallisation trials in LCP, the stabilised receptor (NxStaR) was first fused with *Clostridium pasteurianum* rubredoxin inserted into ICL3[60], as reported previously for the crystal structure determination of the apelin receptor in complex with the synthetic peptide agonist AMG3054[31]. It could be solubilised and purified using the soft detergent dodecyl-β-D-maltopyranoside, in combination with cholesteryl hemi-succinate, in yield and homogeneity amenable to crystallisation experiments. This construct produced small crystals and more sizeable crystallisation hits were optimised by replacing the rubredoxin fusion with the *E. coli* soluble cytochrome b₅₆₂RIL domain inserted between residues 228 and 235, and by carrying out purification and crystallisation experiments in the presence of CMF-019 (see below).

The apelin receptor NxStaR bRIL baculovirus construct was generated using the bac-to-bac system (ThermoFisher Scientific), as per the manufacturer's instructions. For expression in insect cells, *Spodoptera frugiperda* (Sf21) cells were grown at 27 °C with shaking at 135 rpm in 2.5 L total suspension of ESF921 medium (Expression Systems) supplemented with 10% (v/v) foetal bovine serum (Sigma-Aldrich) and 1% (v/v) of a Penicillin-Streptomycin mixture (PAA Laboratories) in 5 L Thomson flasks. Typically, cells were infected at a density of 2.5 to 3.0 × 10⁶ cells/ml with the recombinant virus at an approximate multiplicity of infection of 1, and grown further for 48 h. Cells were then harvested by centrifugation at 3500 rpm at 4 °C, washed in PBS supplemented with Complete EDTA-free protease inhibitor cocktail tablets (Roche), and harvested again by centrifugation at 3500 rpm at 4 °C, and finally stored as pellets at −80 °C.

For membrane preparation, one cell pellet corresponding to 5 L of culture was thawed in water at room temperature for 1 h and resuspended with 300 mL of 25 mM HEPES-NaOH, pH 7.5, 20 mM KCl, 10 mM MgCl₂, and Complete EDTA-free protease inhibitor cocktail tablets (Roche), and stirred for 15 min to remove clumps. Cells were then disrupted in one pass through a microfluidizer at ~15,000 psi (processor M-110L Pneumatic, Microfluidics) cooled with ice. Membranes were pelleted by centrifugation at 200,000 × $g$ at 4 °C for 45 min, and resuspended using a glass dounce homogenizer in lysis buffer supplemented with 1 M NaCl, and centrifuged again at 200,000 × $g$ for 45 min at 4 °C. After removal of the supernatant, membranes were resuspended using a glass dounce homogenizer again in lysis buffer (~150 mL final volume) and stored at −80 °C.

Prior to solubilisation, membranes were incubated with CMF-019 and, subsequently, with iodoacetamide. For solubilisation, the membrane solution was supplemented with buffer including n-Dodecyl-β-D-maltopyranoside (Anatrace) and cholesteryl hemisuccinate (Sigma). For purification, the solubilisation slurry was cleared by centrifugation and the supernatant was loaded onto Strep-Tactin Superflow resin (QIAGEN). After washing the resin, the protein was eluted using D-desthiobiotin (Sigma-Aldrich), and the fractions containing the apelin receptor NxStaR bRIL protein were pooled and concentrated on a 100 kDa concentrator for buffer exchange in buffer without D-desthiobiotin using a PD-10 column (Cytiva). The apelin receptor NxStaR bRIL protein pool was then supplemented with 3C PreScission Protease (Cytiva) and PNGase F protein (New England Biolabs) to

remove the C-terminal purification tag and potential N-glycosylation, respectively. The apelin receptor NxStaR bRIL protein was further purified by incubation with Ni-NTA agarose (QIAGEN) and GST (Cytiva) resins to capture the C-terminal tags and the enzymes. After removal of the affinity resin beads, the protein was further concentrated and spun at $200,000 \times g$ for further analysis on size exclusion chromatography and crystallization experiments.

## Apelin receptor NxStaR crystallisation, structure determination, and refinement

Crystallisation experiments were carried out in LCP at 20 °C. Monoolein (Nu-Chek), supplemented with cholesterol (Sigma) (ratio of 9:1 w/w) and 100 μM CMF-019, was mixed with the protein at a concentration of ~70 mg/mL (based on extinction coefficient and molecular weight of construct) using the twin-syringe method at a ratio of 2:3 (v/v). Boli were dispensed onto Laminex glass bases (Molecular Dimensions Ltd) and overlaid with 800 nL of crystallization solution using a Mosquito LCP crystallization robot (STP Labtech), and sealed using Laminex film covers (Molecular Dimensions Ltd). Rod-shaped crystals used for data collection grew at 20 °C for ~5–6 days in conditions consisting of 25 to 35% polyethylene glycol (PEG) 400, 100 mM sodium citrate, pH 4.75, and 150 mM of ammonium phosphate, potassium phosphate, or sodium acetate, harvested in their mother liquor with a 5% higher PEG 400 concentration, and flash-frozen in liquid nitrogen.

X-ray diffraction data frames (0.25°/frame; ~400 frames per crystal) were collected on a Pilatus 6M detector at beamline I24 (Diamond Light Source). Data from individual crystals were integrated using XDS[61], combined using POINTLESS[62] within the CCP4 program suite[63], and further scaled together using AIMLESS, and corrected for anisotropy using NxSTARANISO within aP_scale[64] (Global Phasing Limited, Cambridge, UK).

The structure of the apelin receptor NxStaR bRIL-CMF-019 complex was determined by molecular replacement (MR) with Phaser[65], using the previously reported structure of the apelin receptor in complex with AMG3054 peptide[31] as the search model (PDB code: 5VBL). Model refinement was performed first using phenix.refine[66] and further using BUSTER[67] (Global Phasing Limited, Cambridge, UK), including TLS refinement for two groups corresponding to the apelin receptor and the bRIL fusion. The final protein structure includes the bRIL fusion in ICL3 and apelin receptor residues 29 to 324, but excluding residues 55 to 61 in ICL1, and 135 to 139 in ICL2. Structure panels were generated using PyMOL. Crystallographic X-ray data collection & structure refinement statistics are provided in the Supplemtary Information (Table S4).

## Surface plasmon resonance

SPR experiments investigating pH dependence of peptide and compound binding were performed using a Biacore 8K+ instrument (Cytiva) and experiments investigating binding of Apelin 13 isoforms were performed on a Biacore T200 instrument (Cytiva). Both were equipped with a NTA sensor chip (Cytiva). Apelin receptors and a negative control were immobilised at 25 °C by a combination of Ni-NTA capture and amine coupling. The buffer used comprised 0.01 M HEPES, pH 7.4, 0.15 M NaCl, 0.02% DDM. 0.5 M EDTA pH 8.0 was injected for 60 s at 5 μL/min followed by a 60 s injection of run buffer containing 0.5 mM $NiCl_2$. The surface was activated with a 1:1 mixture of 0.1 M NHS and 0.5 M EDC at a flow rate of 10 μL/min for 420 s. The receptors were injected at a concentration of 1 μM at 5 μL/min until a surface response of 5000 RU was achieved. The surface was equilibrated for 12 h in buffer before commencing experiments.

Ligand binding kinetics were analysed at 10 °C and had a data collection rate of 10 Hz. The buffer used for measuring peptide isoform binding was 0.01 M HEPES, pH 7.5, 0.15 M NaCl, 0.02% DDM, 5% DMSO. The analyses were run in single-cycle format with five concentration points in a twofold dilution series between 6.25 nM and

100 nM for Apelin-13 and Apelin-13 ΔC2. The concentration range was between 62.5 nM and 1 μM for all other isoforms. The association phase was 120 s for each concentration and the dissociation phase was 600 s. The flow rate was 50 μL/min.

Compound and peptide binding was measured in pH 7.5 buffer previously described and in buffer at pH 6.0 comprising of 0.025 M MES, pH 6.0, 0.15 M NaCl, 0.02% DDM, 5% DMSO. The analyses were run in single-cycle format with four concentration points in a twofold dilution series. The concentration range was 25–200 nM for NXE'992, NXE'145 and NXE'870 and 6.25–50 nM for NXE'686. The association phase was 60 s for each concentration and the dissociation phase was 300 s. There were 4 blank cycles between compound injections.

The data were processed using Biacore T200 Evaluation Software (Version 2.0, Cytiva) and Biacore Insight Evaluation Software (Version 5.0, Cytiva). Sensorgrams were reference and blank subtracted before fitting with 1:1 binding model which accounted for drift, bulk shift and mass transport. Sensorgram images were prepared using Microsoft Excel.

## Human embryonic stem cell culture, differentiation, and purification

Pluripotent, undifferentiated H9 human embryonic stem cells (hESCs, WA09 from WiCell) were maintained as culture colonies on gelatin/MEF media (0.1% gelatin from porcine skin, advanced DMEM/F12, 10% Foetal Bovine Serum (FBS), 1% L-Glutamine, 100 μM β-mercaptoethanol, 100 μ/mL penicillin-streptomycin) coated 6-well plates in chemically defined medium (CDM-BSA: IMDM/F12 (1:1), 15 μg/mL transferrin, 7 μg/mL insulin, 450 mM monothioglycerol, 1% chemically defined concentrated lipids, 5 mg/mL BSA, 100 μ/mL penicillin-streptomycin), supplemented with fibroblast growth factor 2 (FGF2, 12 ng/mL) and Activin-A (10 ng/mL) (maintenance media) with daily media refreshment. Similar protocols have been used previously[51].

For differentiation, cells were plated at ~80,000/well in CDM-BSA supplemented with FGF2 (12 ng/mL), Activin-A (30 ng/mL), and Rho-associated protein kinase inhibitor (ROCKi, 10 μM) on Matrigel-coated 6-well plates. Following incubation at 37 °C for 4 h, mesoderm initiation was started. Cells were treated with 2 mL/well CDM-BSA supplemented with FGF2 (20 ng/mL), Ly294002 (phosphoinositide 3-kinase inhibitor, 10 μM), Activin-A (50 ng/mL) and Bone Morphogenetic Protein 4 (BMP4, 10 ng/mL). Following a further incubation at 37 °C for 42 h, media was removed, and cells washed with PBS, before covering with 2 mL/well CDM-BSA supplemented with FGF2 (8 ng/mL), BMP4 (10 ng/mL), retinoic acid (1 μM), and the WNT signalling pathway inhibitor IWR1-endo (1 ng/mL). After 48 h, media was refreshed, and after a further 48 h, cells were washed with PBS and media was changed to 2 mL/well CDM-BSA supplemented with FGF2 (8 ng/ml) and BMP4 (10 ng/ml).

After 48 h, media was removed, and cells washed with PBS, before covering with 2 mL/well CDM-BSA. Differentiated cardiomyocytes (hESC-CMs) were maintained in CDM-BSA media, with media changes every other day until spontaneous contraction was observed.

For purification, cardiomyocytes were split and re-plated in Matrigel-coated 6-well plates, and covered with CDM-BSA supplemented with ROCKi (10 μM) to promote survival. Cells were incubated overnight to allow adherence and recovery before transferring to lactate selection media (DMEM no glucose, no sodium pyruvate, 1X MEM Non-Essential Amino Acids, 4 mM sodium L-lactate dissolved in HEPES) for 72 h, with one media refresh, in order to generate a pure population of cardiomyocytes. After 72 h, media was changed to CDM-BSA, and refreshed every second day.

## Base editing to generate R/H168[4.64] variant apelin receptor hESC and hESC-CMs

Base editing was used, as described previously[68,69], to generate hESCs and hESC-CMs carrying the R/H168[4.64] apelin receptor variant. Here, a

cytosine base editor was used, in combination with custom guide RNAs (gRNAs), to induce a change in sequence from G-C to A-T, resulting in a substitution mutation of an arginine to a histidine at apelin receptor amino acid position 168.

To design the gRNA, a 1 kb genomic target sequence was input and a ranked list of potential 20 bp gRNAs, based on computationally predicted on- and off-target effects was generated (https://www. benchling.com/). All gRNAs identified directly preceded the PAM motif 5'-NGG, where N represents any base. This PAM sequence is specific for the SpCas9 ortholog and the PAM sequence can occur in the positive or negative strand. For cloning gRNA into the expression plasmid (pGL3-U6-sgRNA-PGK-puromycin plasmid), the vector was cut with BsaI type II restriction enzyme, creating non-compatible sticky ends for the ligation of gRNA. For plasmid preparation, vector was expanded by plating on LB Agar plates containing ampicillin (100 μg/mL), followed by liquid culture in LB broth containing ampicillin (100 μg/ mL). Plasmids were isolated using the Plasmid Plus Midi Kit (QIAGEN), as per the manufacturer's instructions, and NanoDropped to determine DNA concentration.

The gRNA oligonucleotides were then phosphorylated and annealed to form double stranded fragments. Reaction mixture containing 1 μL of top and bottom oligonucleotide (100 μM), 1 μL of T4 Ligase 10X Buffer, 1 μL T4 Polynucleotide Kinase and 6 μL nuclease free water were incubated at 37 °C for 30 min, followed by 95 °C for 5 min, and then temperature ramped down to 25 °C at 5 °C per minute. Annealed R/H168 oligonucleotides were then diluted 1:200 in nuclease free water and reactions set up for ligation of oligonucleotide into the gRNA expression vector (pGL3-U6-sgRNA-PGK-puromycin). Reaction mix consisted of 2 μL diluted annealed oligonucleotides, 100 ng vector, 2 μL Tango Buffer (10X), 1 μL 10 mM DTT, 1 μL 10 mM ATP, 1 μL BsaI FastDigest restriction enzyme, 1 μL T4 Ligase and made up to 20 μL with nuclease free water. Samples were incubated for 6 cycles of 37 °C for 5 min, followed by 21 °C for 5 min.

To remove residual linearised DNA fragments, 11 μL of ligation reaction was mixed with 1.5 μL 10X PlasmidSafe Buffer, 1.5 μL 10 mM ATP and 1 μl PlasmidSafe ATP-dependent DNase, and incubated at 37 °C for 30 min, followed by 70 °C for a further 30 min. Next, 2 μL of the treated ligation reaction was transformed into 25 μL of α-Select Gold Efficiency Chemically Competent Cells and plated on LB Agar + ampicillin (100 μg/mL) plates. Single colonies were selected for liquid starter culture and plasmid DNA isolated using the GenElute Plasmid Miniprep Kit and analysed by Sanger Sequencing (Source Bioscience) using the U6-FOR primer (GACTATCATATGCTTACCGT) to verify presence of gRNA. Positive clones were then regrown in 50 mL liquid culture overnight and plasmid isolated using the Plasmid Plus Midi Kit with an elution volume of 100 μL nuclease free water and concentration determined by NanoDrop.

For nucleofection, hESCs were treated with BE4max base editor vector plus gRNA vector, using the Amaxa 4D Nucleofector (Lonza) and P3 Primary Cell 4D-Nucleofector X Kit, as per manufacturer's instructions. To prepare hESCs for nucleofection, cells were incubated in E8 complete media (DMEM/F12, insulin/transferrin/selenium at 20/ 11/13.4 mg/mL, 0.05% sodium bicarbonate, 7 μM L-Ascorbic acid 2-phosphate, 100 μ/mL penicillin-streptomycin), containing ROCKi (10 μM) for 1 h. For each nucleofection reaction, $1 \times 10^6$ cells/reaction were added to 100 μL Nucleofector solution.

For the base editor reaction, the two vectors were added to Nucleofector solution at a ratio of 1:3 gRNA:base editor, with a total DNA content of 8 μg (2 μg gRNA vector, 6 μg base editor). A control reaction was also set up, with 2 μg of pmaxGFP vector added to Nucleofector solution, in order to visualise successful nucleofection. Each reaction was then transferred to a nucleofector cuvette and placed into the Amaxa 4D Nucleofector, and pulse program CA137 used. Cells were then allowed to recover for 5 min, before adding 500 μL E8 complete media supplemented with CloneR (diluted 1:10,

Stem Cell Technologies) and leaving for a further 5 min. Using the provided plastic transfer pipette, the entire cuvette contents were added to 12 mL of E8 complete +CloneR and plated onto 10 cm dishes. Plates were transferred to 37 °C and incubated overnight to allow attachment. After 24 h, media was changed to E8 complete +puromycin (1 μg/mL), which was maintained for 48 h with daily refresh, before transferring to normal E8 complete. Cells were cultured until visible colonies were present. Resistant colonies were selected manually for expansion and allowed to grow clonally.

To genotype positive colonies, genotyping primers were designed for the *APLNR* nucleotide targeted for genetic manipulation. Sequences ~1 kb up- and down-stream of the residue of interest were input into Primer BLAST online tool and primer pairs generating products with length ~1.5 kb with minimal off target binding and amplification selected. Annealing temperature was determined for the selected primer pair using the New England BioLabs Tm Calculator and performing gradient PCR. Clonal cells were collected and pelleted by centrifugation. Genomic DNA extraction was performed on pelleted cells using the GenElute Mammalian Genomic DNA Miniprep Kit. DNA concentration was determined by NanoDrop and 100 ng of genomic DNA used for each genotyping reaction. The Q5 Hot Start High Fidelity DNA Polymerase kit was used, in combination with the designed sequencing primers. For each reaction, 5 μL was taken for gel electrophoresis to check for appearance of a band at the appropriate size. The remaining 20 μL of reactions producing bands at the predicted size were then PCR purified, NanoDropped and sent for Sanger Sequencing (Source Bioscience). Positive clones were then expanded in culture and differentiated to hESC-CMs for use in further assays.

## Characterisation of R/H168[4.64] hESCs/hESC-CMs in vitro

To assess apelin receptor mRNA expression in R/H168[4.64] variant hESCs/hESC-CMs, RNA extraction was performed using the GenElute Total RNA Purification Kit. Briefly, cells were lysed in 350 μL of RNA lysis buffer and RNA was precipitated with an equal volume of 70% ethanol. Samples were transferred to GenElute Columns for RNA binding, washing, and elution. Samples were eluted in 30 μL Nuclease Free Water, with RNA concentration determined using a NanoDrop 1000 (ThermoFisher). cDNA was produced from 1 μg of RNA using the Promega Reverse Transcription System in a 20 μL reaction, as per the manufacturer's recommendation. To 1 μg RNA, 1 μL of each of Random Primers and Oligo(dT)15 primers were added, made up to 11.9 μL with nuclease free water and incubated at 70 °C for 10 min. A mastermix containing 4 μL $MgCl_2$ (25 mM), 2 μL Reverse Transcription 10X Buffer, 2 μL dNTPs (10 mM), 0.5 μL Recombinant RNasin Ribonuclease Inhibitor and 0.6 μL AMV Reverse Transcriptase per sample was made and 9.1 μL added to each. Samples were then run on a thermocycler. Quantitative real-time polymerase chain reaction (qRT-PCR) was performed for 45 cycles using the ABI 7500 Real-Time PCR System (Applied Biosystems) to analyse mRNA expression using 96-Well TaqMan Gene Expression Assays or SYBR Green Based Assays. Human 18S rRNA or GAPDH were used as house-keeping genes, owing to their stable expression level across cells used. Relative expression was normalised to housekeeping gene expression using the 2(-ΔCT) or the 2(-ΔΔCT) method[70]. CT values over 36 were excluded as non-specific amplicon.

For protein expression, wild-type and R/H168[4.64] hESC-CMs were harvested and plated at ~25,000/well in CellCarrier-96 Ultra Plates (PerkinElmer). To detect apelin receptor using immunocytochemistry, cells were fixed with 4% formaldehyde for 3 min at room temperature. Cells were washed with PBS and blocked with 5% donkey sera in PBS for 1 h. Cells were then treated with primary apelin receptor antibody (Sigma-Aldrich, SAB2700205, 1:50) in PBS containing 3% donkey serum and 1% Tween for 24 h at 4 °C. Subsequently, cells were washed, before treatment with the secondary antibody (abcam, ab150073, 1:200) in PBS containing 3% donkey serum, 1% Tween, and 1:10,000 DAPI

nuclear dye for 1 h at room temperature. For fluorescent ligand (apelin647) binding, cells were treated as described for CHO-K1 cells above. Cells were also imaged as described above using the Opera Phenix High Content Screening System (PerkinElmer).

Saturation radioligand binding, using [125I]-apelin-13, was performed in hESC and hESC-CM membrane preparations as described above for CHO-K1 cells.

The differentiation efficiency of hESCs harbouring the R/H168[4,64] variant into hESC-CMs was determined using flow cytometry, as described previously[51]. Populations of cardiomyocytes were harvested and pelleted by centrifugation at $300 \times g$ for 3 min. Pellets were resuspended in PBS supplemented with 0.1% BSA and 2 mM EDTA (PBE), with CD90 (Thy-1) Monoclonal Antibody directly conjugated to PE diluted at 1:50, for 1 h at 4 °C. Cells were then washed with PBE and resuspended in Fixation/Solubilisation solution (BD Cytofix/Cytoperm Fixation/Permeabilization Kit, Biosciences) for 20 min at 4 °C. Following incubation, cells were washed using 1X BD Perm/Wash Buffer (Biosciences) and then resuspended in 1X BD Perm/Wash Buffer containing directly conjugated Anti-Cardiac Troponin T-APC antibody diluted at 1:50 and incubated for 2 h at 4 °C. Cells were then washed in 1X BD Perm/Wash Buffer, resuspended in PBE and transferred to flow tubes. Samples were run on the LSRFortessa Cell Analyzer (BD Biosciences) and analysis performed using FlowJo v10.8.1 software (Fig. S13).

Apelin peptide secretion was determined via sandwich ELISA, using the apelin-12 (Human, Rat, Mouse, Bovine) EIA kit (Phoenix), as per the manufacturer's instructions. In brief, supernatant samples were added to the secondary antibody coated wells. Primary antibody directed against apelin was added, along with biotinylated apelin peptide, and incubated for 2 h at room temperature with orbital shaking (300 rpm). Wells were washed and blot dried, then streptavidin-conjugated horseradish peroxidase (SA-HRP) was added and incubated for 1 h with orbital shaking. Wells were then washed and dried, and substrate added for 1 h with orbital shaking. The reaction was stopped by the addition of 2 N HCl and absorbance measured at 450 nm using a FLUOstar Omega Microplate Reader. Absorbance is inversely proportional to the concentration of peptide in the sample. Concentrations were determined via extrapolation of a standard curve of known concentrations of BSA.

To assess voltage sensing in hESC-CMs, cells were loaded with FluoVolt Membrane Potential voltage sensitive dye (ThermoFisher, F10488) as per the manufacturer's instructions. FluoVolt was diluted 1:1000 in 1X Tyrode's solution supplemented with glucose (10X Tyrode's Solution diluted with mH2O, 5 mM glucose, pH 7.4), along with PowerLoad solution diluted 1:100. For hESC-CMs in 6-well plates, media was aspirated and replaced with 2 mL/well diluted FluoVolt solution and incubated for 30 min at 37 °C. FluoVolt was then aspirated and replaced with 3 mL/well Tyrode's solution. Cells were imaged using an Axio Observer A1 Inverted Phase Contrast Fluorescence Microscope with LabCam adaptor mounted, and videos recorded in slow motion using an iPhone 7. Cells were paced at 1 Hz, 1.5 Hz and 2 Hz using the C-Pace EM fitted with 6-well plate adaptor (IonOptix). Generated videos were loaded into a custom MATLAB (R2021a) code designed to extract values for time-to-peak and decay time in groups of cells selected as visibly contracting.

### Pharmacological characterisation of peptide agonists

Affinities were determined as described above; see Radioligand Binding section. cAMP (cAMP Hunter™ eXpress AGTRL1 CHO-K1 GPCR Assay; 95-0147E2CP2M) and β-arrestin recruitment (PathHunter® eXpress AGTRL1 CHO-K1 β-Arrestin-1 GPCR Assay; 93-1050E2CP2M) assays were carried out in cells expressing the human apelin receptor according to the manufacturer's protocol to obtain values of $pD_2$. Cardiovascular actions of NXE'065 (30,100 and 150 μg/kg) and NXE'515 (100 and 150 μg/kg) in anaesthetised rat were determined for CMF-

019[30]. Animal experiments were performed in accordance with guidelines from the local ethics committee (University of Cambridge) and the Home Office (UK) under the Scientific Procedures Act (1986).

### Diagrams and schematics

Schema were created with BioRender.com. GPCR snake plots were generated using the GPCRdb server[37]. Chemical structures were drawn using ChemDraw (RRID:SCR_016768).

### Statistical analysis

Data are expressed as mean ± standard deviation (SD). Raw data were handled using Microsoft Excel. Graphical presentation and statistical tests were performed using GraphPad Prism version 6.07. For saturation radioligand binding experiments, data were analysed using the EBDA and LIGAND components of the KELL (Kinetic, EBDA, Ligand, Lowry) software package (Biosoft). For competition binding experiments, Ki values were determined using Cheng-Prusoff methodology. For in vitro GPCR assays, $EC_{50}$, $pD_2$, and $E_{max}$ values were calculated using GraphPad Prism version 6.07. Statistical tests are indicated in figure legends where used, as are experimental $n$ numbers. A $p$-value of <0.05 was determined as significant.

### Reporting summary

Further information on research design is available in the Nature Portfolio Reporting Summary linked to this article.

## Data availability

Source data are provided with this paper as Source Data file. The crystal structure of the stabilized Apelin receptor (NxStaR®) and in complex with CMF-019 described in this study has been deposited at the Protein Data Bank (PDB) under the accession code: https://www.rcsb.org/structure/8S4D. Source data are provided with this paper.

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

## Acknowledgements

This work was supported by: Nxera Pharma ORBIT Research Grant (A.P.D., J.J.M.); Wellcome Trust Programme in Metabolic and Cardiovascular Disease (203814/Z/16/Z, T.L.W.); British Heart Foundation (FS/17/61/33473, A.P.D., R.G.C.M; TG/18/4/33770, A.P.D., J.J.M., R.E.K; FS/18/46/33663, S.S., RM/17/2/33380, M.T.C.); and Cambridge Biomedical Research Centre Biomedical Resources Grant (University of Cambridge, Cardiovascular Theme, RG64226). S.G. was supported by the Cambridge BHF Centre of Research Excellence (RE/18/1/34212). We thank NIHR BioResource volunteers for their participation, and gratefully acknowledge NIHR BioResource centres, NHS Trusts and staff for their contribution. We thank the National Institute for Health and Care Research, NHS Blood and Transplant, and Health Data Research UK as part of the Digital Innovation Hub Programme. We thank the Wellcome-MRC Institute of Metabolic Science Metabolic Research Laboratories Imaging Core for technical support and facilities. The views expressed are those of the author(s) and not necessarily those of the NHS, the NIHR or the Department of Health and Social Care.

## Author contributions

T.L.W. and G.V. designed and carried out experiments, performed data analysis, and wrote and revised the manuscript. R.E.K., H.C., N.S., R.G.C.M., and J.B. designed and carried out experiments, and performed data analysis. G.V., B.B., O.S., M.S., A.Z., and C.d.G. designed and carried out structural experiments and analysis. S.G. performed analysis on genomic data and contributed to writing and review of the manuscript. J.J.M. designed experiments, performed data analysis, and contributed to writing and review of the manuscript. S.S., J.J.M., A.J.H.B., and A.P.D. designed and supervised experiments, performed data analysis, and contributed grant support and facilities. All authors contributed to the writing and/or review of the manuscript.

## Competing interests

G.V., H.C., B.B., N.S., O.S., J.B., M.S., A.Z., and A.J.H.B. are employees of Nxera Pharma UK Limited (Sosei Heptares) and C.d.G. and S.M. are former employees. A.P.D. and J.J.M. were in receipt of a Nxera Pharma UK Limited ORBIT Research Grant. S.G. is co-chair of the International Consortium for Genetic Studies in International Consortium for Genetic Studies in Pulmonary (Arterial) Hypertension. All other authors declare no competing interests.
