## [Peer review file · Nature Communications]

Structural and functional determination of peptide versus small molecule ligand binding at the apelin receptor

Corresponding Author: Professor Anthony Davenport

Version 0:

Reviewer comments:

Reviewer #1

(Remarks to the Author)

The manuscript by the authors presents findings on apelin receptor variants identified from the UK Genomics England 100,000 Genomes Project, highlighting the significance of residues T892.64 and R1684.64. The crystal structure bound to the G protein signaling-biased small molecule agonist, CMF-019, was also determined.

One notable aspect of the small molecule ligand discussed in the article is its functional selectivity toward G protein. However, the discussion and structural explanations regarding this biased signaling lack support from functional data, thereby diminishing the significance of the findings. Additionally, a recent study (PMID: 38428423) reported the cryo-EM structure of the apelin receptor bound to the same ligand (CMF-019), presenting significant advancements in the field. While the focus of this paper is primarily on ligand binding, a more comprehensive functional analysis to assess the downstream effects of the ligand should be conducted. And if possible, I would suggest examining it in the clinically relevant hESC-CM models.

Comments and Suggestions:

1. Validation of ligand binding using radioligand binding assays focused solely on apelin's binding, neglecting ELA's binding. Complementary downstream functional assays such as cAMP accumulation or β -arrestin recruitment assays could enhance the validity of the findings. Similarly, for testing the function of apelin receptor variants, the dynamic mass redistribution assay may not provide sufficient strength. Therefore, supplementing with downstream signaling functional assays is highly recommended.
2. Correct the captions for Fig 1b and 1c, as they appear to be reversed.
3. While we have confidence in the structure prediction abilities of AlphaFold2, considering the existence of reported cryo-EM structures for ELA- and apelin-bound APJ (PMID 3581787 and PMID: 38428423), the authors should conduct additional analysis on these reported structures to enhance the accuracy of the results.
4. Address labeling errors in Fig. S2, including mislabeling of AMG3054 as AMG3096 and misspelling of SdAb.
5. Correct the error in line 472 where V/L381.42 was mistakenly written as V/L381.49.
6. Since the apelin receptor StaR was modified with specific mutations, it is advisable to perform functional assays on these mutations as well to assess their potential impact on the structural conformation.
7. There are labeling errors related to CMF-019 in Fig. 5, including variations like CMF019 and CMF-019. Furthermore, the residue labeling in the lower panel of Fig. 5b (and Fig. 4b) appears unclear and disorganized.
8. The correspondence between ligands and receptors is not clearly visible in the Fig. 6. Please ensure that this is clearly indicated for better clarity.
9. It is advisable to include the effects of small molecules and peptides on the R/H1684.64 apelin receptor variant in the

section of "The R/H168.64 apelin receptor variant induces a detrimental phenotype in hESC-CMs" to elucidate the distinct roles played by different types of ligands.

10. The Introduction and Discussion sections could benefit from more focused descriptions and emphasis on key points.

Reviewer #2

(Remarks to the Author)

The authors describe a structural and functional study of the apelin receptor, a GPCR that binds to apelin and Elabela/Toddler (ELA) and regulates cardiovascular development and function. Analysis of apelin receptor variants from the human genome data resource, in combination with AlphaFold2 modelling, identified T89 and R168 as an important residue binding to the ELA and to the C-termini of both peptides, respectively. Introducing an R168H variant into human stem cell-derived cardiomyocytes (hESC-CM) impaired ligand-receptor binding, differentiation and cellular function. In addition, their newly obtained crystal structure of apelin receptor bound to CMF-019, a G protein-biased agonist, revealed a deeper binding mode compared with peptides at lipophilic pockets between transmembrane helices. They claim that analysis of human genetic variation data is useful to fast identify and characterize key sites that regulate receptor-ligand engagement.

It is a unique approach to search naturally occurring variant of human apelin receptor in the large-scale genome data resource. However, the significant impacts on the fields of their identification of two key amino acid residues (T89 and R168) for ligand-receptor binding is not well convincing for this reviewer. Despite large-scale genome samples, nobody knows that the detection of variants or key residues in 100,000 samples is comprehensive or saturated, and the experiments with hESC-CM did not clarify the physiological relevance of identified variants (T/M89 and R/H168). Overall, their proposal to use human genetic variation to fast identify and characterize the key sites in receptors was unfortunately not well supported by their data. Several critiques are described below.

Major comments

1. To support their proposal, the power of human genetic variation resource to detect key sites needs to be determined by validating known or predicated important sites of the apelin receptor gene combined with structural analysis. Also, haplotype map of the apelin receptor gene would be useful to predict the probable occurrence and sites of variants. Mathematical simulation for detectability of important sites might be also necessary to determine the sample size of genome sequence data.

2. The observation that small molecule agonists bind deeper in the orthosteric pocket of the receptor than peptide ligands ELA and apelin is interesting. The R168 amino acid appeared as amino acid residues interacting with both peptides and small molecules in the list of Figure 6, but the R/H168 variant is not crucial for small molecule agonist to bind the receptor. It would be necessary to investigate which residues are important for small molecule-receptor binding. In addition, since ligand-receptor interaction is a matter in this study, the structure of ligands and small molecule agonists would be necessary to provide for better clarification of similarity and difference in structure of peptides and small molecules.

3. The T/M89 variant narrows binding pocket for ELA, resulting in reduced binding affinity to ELA-11, whereas R/H168 variant diminished both peptide binding to the receptor. Does T/M89 variant exert any phenotypes in humans/mice or in hESC-CM? If any pathological and physiological relevance of this variant should be investigated.

4. Does their proposed strategy contribute to development of better G-protein-biased agonist? Very recently, Wang, et al. reported that they developed a claiming better G-protein-biased agonist than CMF-019 by structure-based designing (Cell. 2024 Mar 14;187(6):1460-1475).

5. For the assessment of physiological relevance of apelin receptor signaling, they used hESC-CM and measured three parameters of cell differentiation efficiency, including TnT expression, apelin secretion and voltage sensing (Figure 8). However, it is important to measure cardiovascular functions in adult animals or isolated cells, such as blood pressure, cardiac hypertrophy, heart contractility in animals, ex vivo isolated blood vessel contractility or in vitro cultured cardiomyocyte hypertrophy. Cell differentiation include complicated process and signaling in addition to G protein signaling and beta-arrestin signaling. It is not suitable to measure the potency of drugs and nor to discriminate G protein signaling and beta-arrestin signaling.

Minor comments

Some of the reference does not match the content of text. There seem errors in numbering the reference.

Reviewer #3

(Remarks to the Author)

This study by Williams et al. is a very well designed, written and executed study. The findings are of great interest to the community, as it provides a platform of using genetic variation to fast-track the identification and characterization of key sites that regulate ligand-receptor engagement. This could have a great impact on drug designing. Genetic variation in GPCRs may contribute to the differentiation of an individual's response to endogenous ligands, thereby giving rise to

pathophysiological conditions and potentially affecting the efficacy of therapeutic agents. The authors characterized coding variants of the apelin receptor by leveraging in vitro pharmacology, molecular modeling, StaR, and gene editing in stem cell-derived cardiomyocytes technologies. They found T892.64 is an important residue in ELA binding, and R1684.64 is a forming extensive interaction with C-termini of both peptides. They also presented a novel apelin receptor crystal structure bound to the G protein-biased agonist CMF-019. Here are some of my comments.

1. The authors give different binding responses of receptor variants to synthetic molecules and peptides; how do these variants affect the clinical efficacy of treatments based on such agonists? Can this study provide a stratified analysis linking variant-specific receptor behavior to potential clinical outcomes?
2. The study notes a significant difference in the binding mode of CMF-019 compared to peptide agonists. In what ways does the binding mode of CMF-019 differ from that of natural peptide agonists, and what implications might these differences have for its function as an agonist?
3. The legends of Figures 1b and c do not correspond to the positions in the figures.

Version 1:

Reviewer comments:

Reviewer #1

(Remarks to the Author)

The authors have addressed my concerns, and I agree with the publication.

Reviewer #2

(Remarks to the Author)

In this revision, the authors toned down the fast-track identification of GPCR variants in the large-scale genome data. Instead, they put more focus on the significance of T89 and R168 residues in apelin receptor regulating receptor-ligand engagement. They repeatedly claim that in the manuscript text and rebuttal letter.

4. Does their proposed strategy contribute to development of better G-protein-biased agonist? Very recently, Wang, et al. reported that they developed a claiming better G-protein-biased agonist than CMF-019 by structure-based designing (Cell. 2024 Mar 14;187(6):1460-1475, PMID: 38428423)

In response to this reviewer's previous comments above, they show the data that half of over 50 peptide apelin agonists are G-protein-biased agonist (Figure S10). The data should be shown in main figures with each peptide sequence and functional validation. The PMID: 38428423 showed that the identified small compounds are much superior to CMF-019 in its potency to improve cardiovascular function. If they say that they identified key sites regulating receptor-ligand engagement, do they develop any improved compounds or peptide ligands better than CMF-019 based on their findings?

In Figure 8, the authors tried to show the functional relevance of apelin receptor variants, this reviewer again points out that the cell differentiation assay of hESC-CM is not suitable for that purpose and that they are still failing to do so. Once again, the authors are advised to measure cardiovascular functions in adult animals or isolated cells, such as blood pressure, cardiac hypertrophy, heart contractility in animals, ex vivo isolated blood vessel contractility or in vitro cultured cardiomyocyte hypertrophy.

Reviewer #3

(Remarks to the Author)

The authors have addressed all of my comments.

Version 2:

Reviewer comments:

Reviewer #2

(Remarks to the Author)

Point 1. Does their proposed strategy contribute to development of better G-protein-biased agonist? Very recently, Wang, et al. reported that they developed a claiming better G-protein-biased agonist than CMF-019 by structure-based designing (Cell. 2024 Mar 14;187(6):1460-1475, PMID: 38428423)

In response to this reviewer's previous comments above, they show the data that half of over 50 peptide apelin agonists are G-protein-biased agonist (Figure S10). The data should be shown in main figures with each peptide sequence and functional validation. The PMID: 38428423 showed that the identified small compounds are much superior to CMF-019 in its potency to improve cardiovascular function. If they say that they identified key sites regulating receptor-ligand engagement,

do they develop any improved compounds or peptide ligands better than CMF-019 based on their findings?

We thank the reviewer for these further comments. Yes, to confirm we have compounds that are better than CMF-019. However, we would highlight again that this was not the purpose of the manuscript but in fact in our previous revision, we added data to the supplementary file showing that we have generated compounds which are more G protein biased (over beta-arrestin) than CMF-019 (Supplementary Figure 10B). We have in this revision included the individual data for the compounds used to generate Supplementary Figure 10.

Our previously published data using the same assays as used in this manuscript (Read et al., 2016, PMID:27475715) showed that CMF-019 was biased towards the G protein pathway over beta-arrestin by three orders of magnitude (2250). In this second revision of the manuscript, we report compounds that are up to four orders of magnitude (for example, NXE'779, 76 200 fold selective for cAMP versus beta-arrestin) in a new Table S5. Please note that Sosei Heptares has now changed to Nxera.Life. The prefix for compounds was 'HTL' and now changed to 'NXE', which has been amended in the submitted files. Table S5 includes competition binding data for our compounds tested against [125I]-apelin in clinically relevant human heart, and comparison of the potency of compounds in the G protein pathway (cAMP) versus beta-arrestin assays to show G protein selectivity. We have also added additional in vivo data from two representative compounds, one biased towards the G protein pathway over beta-arrestin (NEX'065) and one that displays no bias (NEX'515). As previously stated, these data importantly show that both non-biased and G protein-biased agonists produce the same desired short term beneficial actions: increasing cardiac output and decreasing in left ventricular systolic pressure and arterial pressure. This profile of a G protein-biased agonist has also been shown by us in proof of principle studies to be beneficial in the Sugden-hypoxia animal model of pulmonary arterial hypertension.

The main manuscript is already at the maximum of 10 display items (tables and figures); therefore the new Table S5 and Figures S11 have been added to the Supplementary file.

[Comments] The authors provided the new Table S5 and Figures S11. They should mention the compound NXE'779 in the main text, which is 76,200 fold selective for cAMP versus beta-arrestin and is likely superior to CMF-019, as listed in a new Table S5. This is important, because they eventually claim the 'clinical relevance' of this study, and then readers always ask the potency of identified compounds. In addition, the context of NEX'065 and NEX'515 should be described in the discussion section. This would help general readers to understand this study better.

Point 2. In Figure 8, the authors tried to show the functional relevance of apelin receptor variants, this reviewer again points out that the cell differentiation assay of hESC-CM is not suitable for that purpose and that they are still failing to do so. Once again, the authors are advised to measure cardiovascular functions in adult animals or isolated cells, such as blood pressure, cardiac hypertrophy, heart contractility in animals, ex vivo isolated blood vessel contractility or in vitro cultured cardiomyocyte hypertrophy.

We thank the referee for this advice. Naturally occurring variants in the apelin receptor in humans have not been characterised before and the impact of a single amino acid change on functional responses is difficult to predict in G protein coupled receptors. We therefore used CRISPR base editing to introduce the apelin receptor variant into human stem cell derived cardiomyocytes. This is a technically challenging experimental technique but important to carry out in human cells for clinical relevance and as proof of principle that the variant may (may not) affect function.

These cardiomyocytes were used in as an initial screen and we were able to show potential functional significance of the variant in several assays (not just cell differentiation), including voltage sensing studies that showed that the R/H1684.64 hESC-CMs exhibited a prolonged time-to-peak voltage. This suggests an influence on the cardiac action potential in cardiomyocytes. Phase 0 of the cardiac action potential describes the rapid depolarization of the cell membrane voltage, predominantly relating to activation of Na⁺ channels and influx of Na⁺ into the cell. We can speculate for example, that apelin may influence activation of these ion channels. Genetic disorders of ion channels may present with familial forms of arrhythmia, for example SCN5A in Brugada syndrome and some forms of long QT syndrome.

As Referee 2 will appreciate, the request for in vivo data characterising animals, where base editing has been used to introduce this variant, is even more technically challenging and would take many months to generate. We are not able to provide carry out the studies suggested. However, we hope that publication of the manuscript will stimulate further research as to whether apelin receptor variants will have clinical consequences and inform the potential of the apelin signalling pathway in as a target in cardiovascular disease.

[Comments] If they claim the functional relevance of R/H168_4.64, only by showing the results of cell culture experiments of hESC-CMs, they should tone down the 'clinically relevant' cells/settings in the text to simply mentioning 'human' embryonic stem-cell derived cardiomyocytes. If they still want to say 'clinically relevant' cells, they should establish iPS cells from the patients with R/H168_4.64 variant, differentiate it into cardiomyocytes, and compare it with control wild type cells. If the detrimental phenotype is reproducible with the patient cells, they should also conduct CRISPR base-editing the variant cells to correct R/H168_4.64 to wild type. In addition, the derivation of cardiomyocytes was conducted only from one strain of hESCs (WA09 from WiCell). If another hESC strain is used for the same experiment, would the results be reproducible? Furthermore, the detrimental phenotype is so unclear. They describe that both development and GPCR signaling may be important. However, they should clarify whether the phenotype is due to development or GPCR signaling. The authors are strongly advised to conduct RNA sequence analysis and determine how differentiation is impaired. Since TnT positive-differentiated cardiomyocytes are markedly decreased by R/H168_4.64 variant, it is not unlikely that regulation of Na channel is altered in the signaling levels, thereby time-to-peak voltage prolonged. Rather, simply undifferentiated cells are the cause.

Version 3:

Reviewer comments:

Reviewer #2

(Remarks to the Author)

The authors claim that the phenotype of R168H variant-knockin hESC-CM is the same as that of apelin receptor-knockdown hESC-CM in their previous report and thus that it's not necessary to confirm the phenotype-genotype relationship.

Are their apelin receptor-impaired hESC-CMs always the same? Is autocrine loop of apelin ligand – apelin receptor really necessary for maintaining SCN5A expression, instead of ligand-independent activation of apelin receptor? How about involvement of elabela?

When wild type hESCs are treated with an apelin-neutralizing antibody throughout differentiation to hESC-CM, do they recapitulate R168H variant-knockin or apelin receptor-knockdown?

This reviewer does not agree that they confirm the phenotype-genotype relationship for R168H variant.

Version 4:

Reviewer comments:

Reviewer #2

(Remarks to the Author)

The authors addressed this reviewer's concern.

NCOMMS-24-09808 *Structural and functional determination of peptide versus small molecule ligand binding at the apelin receptor* by Thomas L. Williams, Grégory Verdon, Rhoda E. Kuc, Heather Currinn, Brian Bender, Nicolae Solcan, Oliver Schlenker, Robyn G.C. Macrae, Jason Brown, Marco Schütz, Andrei Zhukov, Sanjay Sinha, Chris de Graaf, Stefan Gräf, Janet J. Maguire, Alastair J.H. Brown, and Anthony P. Davenport.

REVIEWER COMMENTS

We thank the referees for their extensive review of the manuscript and for their comments.

We are very pleased that Referee 3 found: *'This study by Williams et al. is a very well designed, written and executed study. The findings are of great interest to the community, as it provides a platform of using genetic variation to fast-track the identification and characterization of key sites that regulate ligand-receptor engagement. This could have a great impact on drug designing'*.

Referee 2: *'It is a unique approach to search naturally occurring variant of human apelin receptor in the large-scale genome data resource'*.

We have extensively revised the manuscript as requested (additions/changes highlighted in yellow) and included new data, with a revised Fig 6, two new figures and one table in the supplementary file. We have responded to all of the comments in the response to referees including with the addition of new data in the manuscript and supplementary file. Where the information requested has already been published (physiological and pharmacological actions of apelin, ELA, our biased compounds MM07 peptide and importantly the small molecule, CMF-019 used to crystallise the receptor) these data have been reproduced for the convenience of the referees in our response, compiled from several different papers. In the manuscript, we have added a summary sentence in the introduction and directed the readers to these relevant papers, within the constraints of a maximum 70 references.

Referees comments in black, our comments in blue and text added to manuscript in yellow highlight.

Reviewer #1 (Remarks to the Author):

The manuscript by the authors presents findings on apelin receptor variants identified from the UK Genomics England 100,000 Genomes Project, highlighting the significance of residues T892.64 and R1684.64. The crystal structure bound to the G protein signaling-biased small molecule agonist, CMF-019, was also determined.

One notable aspect of the small molecule ligand discussed in the article is its functional selectivity toward G protein. However, the discussion and structural explanations regarding this biased signaling lack support from functional data, thereby diminishing the significance of the findings. Additionally, a recent study (PMID: 38428423) reported the cryo-EM structure of the apelin receptor bound to the same ligand (CMF-019), presenting significant advancements in the field. While the focus of this paper is primarily on ligand binding, a more comprehensive functional analysis to assess the downstream effects of the ligand should be conducted. And if possible, I would suggest examining it in the clinically relevant hESC-CM models.

Response

We thank the referee for their thorough reviewing of the manuscript and suggestions. As explained below the referee refers to a recent study (PMID: 38428423) published 29th Feb 2024 reporting on the use of a different technique (cryo-EM structure), whereas we are reporting on the X-ray structure, and we submitted our manuscript a week earlier on 19th Feb 2024. Therefore, their manuscript was not available to us.

The referee states that PMID: 38428423 is a significant advance in the field of biased agonism.

We note the referee indicates discussion and structural explanations regarding this biased signalling lacks support. We were the first to identify the small molecule ligand CMF-019, that we had previously extensively characterised as G protein biased (PMID: 27475715). The availability of this compound resulted in the successful crystallization of the receptor. We have discussed an explanation of structural bias in the discussion:

Another key difference with peptides is the different side chain conformation of Y299^{7.43}, which has been linked to bias. Here, we observed CMF-019 stabilising a rotamer of Y299^{7.43} (Fig. 5a) that is distinct from the rotamer observed in the peptide complex structures (Fig. 5b). The Y299^{7.43} residue is conserved among other class A GPCRs, where it plays an important role as a 'bias switch'. For instance, ligand-induced conformational changes in the corresponding Y291^{7.43} residue in the chemokine receptor CCR1 allosterically regulate β -arrestin activation to drive signalling bias⁴⁹. Additionally, mutagenesis in the AT₁ receptor has previously demonstrated an impairment of a Y/F292^{7.43} mutant to recruit β -arrestin in response to a benchmark β -arrestin-biased ligand⁵⁰. Therefore, binding in this sub-pocket III at the interface between TM6 and TM7 may contribute to G protein-bias in the apelin receptor, and warrants further investigation in order to inform the design of biased drugs that might offer better therapeutic outcomes.

The focus of PMID: 38428423 was to identify key determinants of apelin receptor signalling bias including CMF-019; since this is covered comprehensively, we do not believe additional comments are needed in the manuscript.

We were able to design and synthesise both G protein biased and balanced peptide ligands (new supplementary Figure S10) but understanding the specific SAR and interacting amino acids underpinning this pathways bias was not our primary aim.

We have, responded specifically to Point 3 with a new panel in Fig 6 and added a new Supplementary Figure S4 to discuss and highlight structural comparisons with the recently published apelin peptide and CMF-019 bound cryoEM structures in PMID: 38428423. We have also re-analysed data in order to make these comparisons as accurate as possible. Importantly, PMID: 38428423 specifically concurs with the conclusions in our manuscript regarding the key amino acid residues in the binding pocket of the receptor interacting with the small molecule ligands. This comparison is included in new text in Fig 6 as well as in the results and discussion sections of our revised manuscript.

However, we would like to clarify that a major part of our manuscript is different from PMID: 38428423. We have focussed on understanding the intriguing differences between the two endogenous peptides, apelin and ELA, binding to the receptor, informed by the discovery and characterization of naturally occurring apelin receptor variants for the 100,000 genomes. Furthermore, naturally occurring apelin receptor variants in humans have not previously been described and characterised.

We believe our result complement and extend the results of PMID: 38428423. This is of course in line with the stated objective of Nature Communications as highlighted in a recent editorial entitled 'Strength in numbers' (DOI<https://doi.org/10.1038/s41467-020-17817-x>)

'The robustness of science is best revealed when independent investigations of the same problem arrive at similar conclusions. At Nature Communications, we commit to disregard from our editorial evaluation any competing works that are published while a submission to our journal is under review or under revision by the authors'.

We have also responded in Point 1 to the request for downstream signalling and the other variant in hESC. (Point 9).

Comments and Suggestions:

1. Validation of ligand binding using radioligand binding assays focused solely on apelin's binding, neglecting ELA's binding. Complementary downstream functional assays such as cAMP accumulation or β -arrestin recruitment assays could enhance the validity of the findings. Similarly, for testing the function of

apelin receptor variants, the dynamic mass redistribution assay may not provide sufficient strength. Therefore, supplementing with downstream signaling functional assays is highly recommended.

We thank the referee for these comments.

The binding of ELA has not been neglected. We show in Fig 2g competition binding for unlabelled ELA versus labelled apelin in T/89M variant versus wild type.

[¹²⁵I]-ELA-11 is not commercially available and to date radiolabelled ELA has not been reported by any other group. We have tested custom synthesised [¹²⁵I]-ELA in human tissue labelled at the C or N terminus but neither radioligand displayed specific binding. We therefore used fluorescent apelin and ELA ligands, which have been characterised in-depth previously (Williams et al., 2023), to investigate the variants and only T/M89 showed any discrepancy in binding between the two ligands (Fig 2b).

Validation of ligand binding assays has already been carried out by us (Yang et al., Circulation. 2017 PMID: 28137936) for both [Pyr¹]apelin-13 and all the predicted ELA isoforms in human heart (see figure below). ELA isoforms compete with high affinity for radiolabelled apelin binding so it is not necessary to repeat these experiments.

Figure shows competition binding curves for [Pyr¹]apelin-13 and ELA isoforms in human heart against [¹²⁵I]apelin-13.

Similarly, complementary downstream functional assays have been carried out, by us, for both apelin and ELA and are shown in the table and figure below. We have shown that predicted ELA isoforms and apelin activate downstream signalling pathways with cardiovascular relevance, specifically inhibition of cAMP formation, phosphorylation of ERK1/2, phosphorylation of nitric oxide synthase, β -arrestin recruitment and receptor internalization (Yang et al., Circulation. 2017 PMID: 28137936)

Potency (pD₂) and Efficacy (EMAX) of [Pyr¹]apelin-13, ELA-32, ELA-21, and ELA-11 in cAMP Inhibition, β -Arrestin Recruitment, and Receptor Internalization Assays

Peptide	cAMP		β -Arrestin		Internalization	
	pD ₂	E _{MAX} %	pD ₂	E _{MAX} %	pD ₂	E _{MAX} %
[Pyr ¹]apelin-13	9.74±0.11	100±1	8.10±0.12	100±2	8.87±0.25	100±2
ELA-32	9.49±0.07	101±1	9.05±0.32††	104±4	9.66±0.36†	102±4
ELA-21	9.43±0.16	101±1	9.88±0.29**†††	102±1	9.55±0.08	96±2
ELA-11	9.13±0.21	100±1	7.44±0.18	107±6	8.13±0.56	105±1

Values are mean±standard error of the mean, n=3–5 assays, 2–6 replicates per assay. ELA indicates Elabela/toddler.

**P≤0.01, significantly different from [Pyr¹]apelin-13.

†P≤0.05, ††P≤0.01, †††P≤0.001, significantly different from ELA-11.

Figure 3.

Concentration-response curves in cell-based receptor pharmacology assays. A, Inhibition of forskolin-induced cAMP accumulation. B, Stimulation of β -arrestin recruitment. C, Induction of receptor internalization. [Pyr¹]apelin-13, ELA-32, ELA-21, and ELA-11. Antagonism of [Pyr¹]apelin-13 (D) and ELA-32 (E) by 30 μ M ML221 in the β -arrestin assay. Concentration-response curves to ELA-14 and cyclic ELA-11 in cAMP (F) and β -arrestin (G) assays. Values are mean \pm SEM, n=3 to 5 assays, 2 to 6 replicates per assay. ELA indicates Elabela/toddler; and SEM, standard error of the mean.

Effect of [Pyr¹]Apelin-13 and ELA-32 on levels of protein phosphorylation in PAEC and PASMCS in vitro. Increased phosphorylation levels of (A) ERK1/2 and (B) eNOS in PAECs and ERK1/2 in (C) control and (D) PAH PASMCS following treatment with 0.1% serum (control), [Pyr¹]apelin-13 or ELA-32 (both 100nmol/L). Significantly different from control * $P \leq 0.05$, ** $P \leq 0.01$, one-way ANOVA for repeated measures with Tukey's post-test for multiple comparisons.

In all these assays the effects of apelin and ELA peptides are comparable, supporting no discernible difference in downstream signalling with the endogenous ligands used in our study.

The Referee suggests that the dynamic mass redistribution assay may not provide sufficient strength. However, dynamic mass redistribution assays are a widely used (over 400 references in Pubmed) and generally accepted robust label-free technology to measure ligand-induced changes in the mass of cells and provides an integrated cellular response, comprising multiple pathways and cellular events. We believe DMR is robust for testing the function of apelin receptor variants, particularly when combined with our radioligand binding assays.

The Referee suggested that 'supplementing with downstream signaling functional assays is highly recommended.' We thank the referee for the recommendation but we were not successful in generating additional data in the apelin receptor variants with downstream signalling functional assays that we carried out.

We have tried to test the function of apelin receptor variants in a DiscoverX cAMP assay. The figure below shows the responses to [Pyr¹]apelin-13 (left) and CMF-019 (right).

Provided in the assay kit, CHO-K1 cells stably expressing the apelin receptor (shown in green) responded as expected to stimulation with [Pyr¹]apelin-13 and CMF-019, with G_i coupled G protein activation reducing the levels of forskolin stimulated cAMP accumulation and decreasing the fluorescent signal in relative light units (RLU). CHO-K1 cells transiently transfected in-house with either wild-type (black), T/M89 (red), or R/H168 apelin receptor (purple) did not show an observable response to either ligand in this assay. This is likely due to low transfection efficiency resulting in too few cells expressing the receptor to be able to observably inhibit the stimulation of cAMP by forskolin in all the cells in the population. Essentially, this meant that we did not have an appropriate window to see an effect in this assay using the transiently transfected cells, unlike the DMR which was more sensitive and allowed us to capture a robust response. Overall cell density was similar in all cases as forskolin induced a similar initial response (~150,000 RLU) at the lowest doses of the test ligands. The apelin receptor is a G_i coupled GPCR and this poses challenges in assay development that are mitigated using DMR.

2. Correct the captions for Fig 1b and 1c, as they appear to be reversed.

We thank the referee for this point, changed as requested.

3. While we have confidence in the structure prediction abilities of AlphaFold2, considering the existence of reported cryo-EM structures for ELA- and apelin-bound APJ (PMID 3581787 and PMID: 38428423), the authors should conduct additional analysis on these reported structures to enhance the accuracy of the results.

We thank the referee for this suggestion. We note the referee refers to a recent study (PMID: 38428423) published on 29th Feb reporting on the cryo-EM structure, whereas we are reporting on the X-ray structure submitted a week earlier on 19th Feb and therefore PMID: 38428423 was not available to us.

We have, as requested, updated the manuscript by incorporating a new panel in Fig 6 and a new Supplementary figure S4 on the structural comparisons with the recently published apelin peptide and CMF-019 bound cryo-EM structures in PMID: 38428423. We have added a detailed comparison as highlighted in our revised manuscript as follows:

We further compared our apelin receptor StaR

CMF-019 crystal structure with previously reported apelin receptor-peptide bound experimental structures (Fig. 6). Overall, we find that peptides engage with the receptor via the very large space at the interface between TM7, TM2, TM3, and ECL2, with their C-termini reaching down in the orthosteric pocket at a site lying at the interface between TM3, TM5, TM6, and TM7 (Fig. S2, S6). This part of the orthosteric pocket is lined by the side chains of residues F110^{3.33}, Y264^{6.51}, and K268^{6.55} (Fig. S2b) and corresponds roughly to where the small agonist cores are positioned. In this configuration, the conformation of the ELA F10 side chain across the cryoEM structures, for example, is such that it visits either sub-pocket II or III, and the location of the C-terminal carboxylate overlaps with that of the oleic acid head group. Similarly in the recent

apelin bound cryoEM structure, M11 side chain overlaps with the CMF-019 thiophene moiety, and P12 and F13 locations overlap with that of the oleic acid.

However, striking differences were observed for the small molecule versus peptide binding sites. Notably, the small molecule agonists appeared to reach deeper towards the bottom of the orthosteric site, especially at the TM6-TM7 interface. Across these structures, the most noteworthy side chain rotamer differences observed at the bottom of the orthosteric site were for residues Y35^{1,39}, F78^{2,53}, W85^{2,60}, F110^{3,33}, and Y299^{7,43}, and the small molecule agonists in active conformational states stabilise a rotamer of Y299^{7,43} that is lodged into the interface between TM7 and TM2 (sub-pocket III). Additionally, as mentioned above, the R168^{4,64} side chain shows different positions and conformations, with some positioning the guanidinium group of R168^{4,64} closer to peptide C-termini, so that it interacts with backbone oxygens. In the recent apelin cryoEM structure, such an interaction is only observed with the peptide serine at position 6. Of interest, the R168^{4,64} residue only forms a minor direct interaction with the core of CMF-019, and engaging mostly indirectly via a salt bridge with the oleic acid molecule that was observed to protrude into the binding site (Fig. 4).

4. Address labeling errors in Fig. S2, including mislabeling of AMG3054 as AMG3096 and misspelling of SdAb.

We thank the referee for this point, changed as requested.

5. Correct the error in line 472 where V/L381.42 was mistakenly written as V/L381.49.

We thank the referee for this point, changed as requested.

6. Since the apelin receptor StaR was modified with specific mutations, it is advisable to perform functional assays on these mutations as well to assess their potential impact on the structural conformation.

We thank the referee for this point, we carried out these assays to identify the changes to the apelin receptor StaR had no effect on GFP and radioligand binding. The table below shows data for the Star mutations, and this has been added to the Supplementary data file as new Supplementary Table S3.

StaR mutation	GFP (% WT)	B _{max} (4 °C)	Ratio	Ratio - WT	B _{max} (% WT)	Stability index
T/V87 ^{2,62}	139.68	483.00	0.71	0.28	99.26	1.28
N/A112 ^{3,35}	78.48	705.50	0.77	0.24	121.49	1.24
T/M207 ^{5,44}	118.43	470.50	0.27	-0.08	122.43	0.92
F/L214 ^{5,51}	90.37	408.50	0.65	0.10	108.28	1.10
I/A224 ^{5,61}	89.08	1409.00	0.48	-0.07	373.49	0.93
S/A298 ^{7,42}	99.94	3119.00	0.80	0.27	248.23	1.31

Legend for Supplementary Table 3: Summary of the impact of the selected apelin stabilised receptor (StaR) mutations on expression, radioligand binding, and thermostability. Overall expression was determined using GFP signal detected by PHERAstar microplate reader and reported as a percentage of the wild-type (WT) template. Receptor density (B_{max}) was determined in the presence of the radioligand [³H]-NXE0025870 at 4

°C. Thermostability was measured by incubation at different temperatures and expressed as a ratio (B_{\max} at 4 °C / B_{\max} at higher temperature). B_{\max} was also expressed as a percentage of the WT template. Importantly, a stability index ($1 + (\text{Ratio} - \text{WT})$) greater than 1 is classed as stabilising versus the WT template.

7. There are labeling errors related to CMF-019 in Fig. 5, including variations like CMF019 and CMF-019. Furthermore, the residue labeling in the lower panel of Fig. 5b (and Fig. 4b) appears unclear and disorganized.

We thank the referee for this point, corrections have been made as requested.

8. The correspondence between ligands and receptors is not clearly visible in the Fig. 6. Please ensure that this is clearly indicated for better clarity.

We thank the referee for this point, changed as requested. We have modified with an extra panel to improve clarity. We have added text to explain Figure 6 and additional explanation as follows.

We further compared our apelin receptor StaR CMF-019 crystal structure with previously reported apelin receptor-peptide bound experimental structures (Fig. 6). Overall, peptides engaged with the receptor via the large space at the interface between TM7, TM2, TM3, and ECL2, with their C-termini reaching down in the orthosteric pocket at a site lying at the interface between TM3, TM5, TM6, and TM7 (Fig. S2). This part of the orthosteric pocket is lined, for instance, by the side chains of residues F110^{3.33}, Y264^{6.51}, and K268^{6.55} (Fig. S2). Compared to the small molecule agonist complex structures, this corresponds roughly to where the small agonist cores are positioned. In this configuration, the conformation of the ELA F10 side chain across the cryoEM structures, for example, is such that it visited either sub-pocket II or III, and the location of the C-terminal carboxylate overlapped with that of the oleic acid head group. Similar to the recent apelin bound cryoEM structure, the M11 side chain overlapped with the CMF-019 thiophene moiety, and P12 and F13 locations overlapped with that of the oleic acid.

However, striking differences were observed for the small molecule versus peptide binding sites. Notably, the small molecule agonists appeared to reach deeper towards the bottom of the orthosteric site, especially at the TM6-TM7 interface. Across these structures, the most noteworthy side chain rotamer differences observed at the bottom of the orthosteric site were for residues Y35^{1.39}, F78^{2.53}, W85^{2.60}, F110^{3.33}, and Y299^{7.43}, and the small molecule agonists in active conformational states stabilised a rotamer of Y299^{7.43} that is lodged into the interface between TM7 and TM2 (sub-pocket III). Additionally, as mentioned above, the R168^{4.64} side chain showed different positions and conformations, with some positioning the guanidinium group of R168^{4.64} closer to peptide C-termini, so that it interacted with backbone oxygens. In the recent apelin cryoEM structure, such an interaction is only observed with the peptide S6 sidechain. Of interest, the R168^{4.64} residue only formed a minor direct interaction with the core of CMF-019, engaging mostly indirectly via a salt bridge with the oleic acid molecule that was observed to protrude into the binding site (Fig. 4).

9. It is advisable to include the effects of small molecules and peptides on the R/H1684.64 apelin receptor variant in the section of "The R/H1684.64 apelin receptor variant induces a detrimental phenotype in hESC-CMs" to elucidate the distinct roles played by different types of ligands.

We thank the referee for this advice. We believe that we have already carried out substantive characterization of this variant in CHO cells and in the context of extensive modelling in relation to the amino acid residues in the binding pocket of the receptor interacting with the small molecule ligands as well as the endogenous peptides. Our focus has been the binding of apelin and ELA to the receptor. The apelin receptor is unusual for a GPCR in having two structurally distinct ligands from different peptide families, that can bind with similar high affinity.

10. The Introduction and Discussion sections could benefit from more focused descriptions and emphasis on key points.

We have revised the text to manuscript to give more focus. These changes are indicated throughout the revised manuscript in yellow highlights.

Reviewer #2 (Remarks to the Author):

The authors describe a structural and functional study of the apelin receptor, a GPCR that binds to apelin and Elabela/Toddler (ELA) and regulates cardiovascular development and function. Analysis of apelin receptor variants from the human genome data resource, in combination with AlphaFold2 modelling, identified T89 and R168 as an important residue binding to the ELA and to the C-termini of both peptides, respectively. Introducing an R168H variant into human stem cell-derived cardiomyocytes (hESC-CM) impaired ligand-receptor binding, differentiation and cellular function. In addition, their newly obtained crystal structure of apelin receptor bound to CMF-019, a G protein-biased agonist, revealed a deeper binding mode compared with peptides at lipophilic pockets between transmembrane helices. They claim that analysis of human genetic variation data is useful to fast identify and characterize key sites that regulate receptor-ligand engagement.

It is a unique approach to search naturally occurring variant of human apelin receptor in the large-scale genome data resource. However, the significant impacts on the fields of their identification of two key amino acid residues (T89 and R168) for ligand-receptor binding is not well convincing for this reviewer. Despite large-scale genome samples, nobody knows that the detection of variants or key residues in 100,000 samples is comprehensive or saturated, and the experiments with hESC-CM did not clarify the physiological relevance of identified variants (T/M89 and R/H168). Overall, their proposal to use human genetic variation to fast identify and characterize the key sites in receptors was unfortunately not well supported by their data. Several critiques are described below.

We appreciate the comments about comprehensive or saturated variants but this was not the purpose of the study. It was to identify naturally occurring rare variants in highly conserved regions as a starting point to understand ligand receptor interactions with the two endogenous ligands. These are from structurally distinct families but unusually have evolved to bind the same receptor. We used the T/M89 variant to identify the T89 amino acid as a critical residue for ELA but not apelin binding and therefore function. This has not been described before. Such functionally distinct naturally occurring variants have not be described before. We hope with this explanation, the referee will allow us to retain the following modified sentence:

Overall, the data provide proof-of-principle for using genetic variation to identify key sites regulating receptor-ligand engagement.

Major comments

1. To support their proposal, the power of human genetic variation resource to detect key sites needs to be determined by validating known or predicated important sites of the apelin receptor gene combined with structural analysis. Also, haplotype map of the apelin receptor gene would be useful to predict the probable occurrence and sites of variants. Mathematical simulation for detectability of important sites might be also necessary to determine the sample size of genome sequence data.

We thank the referee for this suggestion about known or predicated important sites of the apelin receptor gene. As we indicate, we are not aware of previously published naturally occurring single point variants in humans. As we state in the methods section (headed Identifying rare sequence variation in the apelin receptor (APLNR) gene) we used SIFT and PolyPhen-2 to identify single point mutations predicted to cause major deleterious changes in the physicochemical properties in the substituted amino acid. As indicated, we focused only on highly conserved amino acids across species, which are more likely to be under selection pressure. A third criterion to select the variants for study was to identify corresponding amino acids (using Ballesteros–Weinstein numbering scheme) critical for binding in other GPCRs based on calculations from the database GPCRdb.

Using the above criteria, we first identified variants in human genome cohort then subsequently selected those that were likely to have physiological effect and might point to a clinical correlation which could be assessed in more detail in future studies. This strategy was used to rapidly identify potentially important binding/functional sites to better understand receptor function and support future drug design strategies by targeting, or even specifically avoiding, certain residues.

We have removed ‘fast-track’ from the abstract

We agree that it is uncertain whether the available population-scale sequencing data is comprehensive or saturated. When we initially prioritized and selected the variants, we used available databases and resources that already included a large number of samples. We have now subsequently re-analysed the occurrence of the variants in the latest versions of gnomAD, noting that version 4.1.0 comprises samples collated from gnomAD and UK BioBank (totalling 807,602 samples), to confirm that they are ultra-rare in these populations.

This new analysis using the most recent databases did not change our original identification of the variants, supporting the power of this human genetic variation resource to detect key sites. We have added a sentence to the Methods section of the manuscript to explain this:

'After initial filtering, subsequent analyses of genetic data in the gnomAD (version 4.1.0) and UK BioBank cohorts, comprising a total of 807,602 samples, confirmed that selected variants remained at an ultra-rare level of occurrence across these datasets.'

In support of the above sentence, the allele frequencies for variants inducing an amino acid change at the selected sites T89 and R168 for example, remain ultra-rare across the gnomAD data set, occurring in less than 4 in 100,000 and 6 in a 100,000 respectively (see Table below).

Variant	Resource	Version	HGVSg	HGVSp	Allele count	Allele number	Allele freq.
T89M	gnomAD	4.1.0	11-57236739-G-A(GRCh38)	p.Thr89Met	34	1613884	2.11E-05
	gnomAD	3.1.2	11-57236739-G-A(GRCh38)	p.Thr89Met	6	152244	3.94E-05
	gnomAD	2.1.1	11-57004213-G-A(GRCh37)	p.Thr89Met	5	251052	1.99E-05
T89L	gnomAD	4.1.0	11-57236739-G-T(GRCh38)	p.Thr89Lys	5	1614002	3.10E-06
	gnomAD	2.1.1	11-57004213-G-T(GRCh37)	p.Thr89Lys	1	251052	3.98E-06
T89R	gnomAD	4.1.0	11-57236739-G-C(GRCh38)	p.Thr89Arg	1	1614002	6.20E-07
	gnomAD	2.1.1	11-57004213-G-C(GRCh37)	p.Thr89Arg	1	251052	3.98E-06
R168H	gnomAD	4.1.0	11-57236502-C-T(GRCh38)	p.Arg168His	34	1614064	2.11E-05
	gnomAD	3.1.2	11-57236502-C-T(GRCh38)	p.Arg168His	9	152192	5.91E-05
	gnomAD	2.1.1	11-57003976-C-T(GRCh37)	p.Arg168His	4	250698	1.60E-05
R168P	gnomAD	4.1.0	11-57236502-C-G(GRCh38)	p.Arg168Pro	2	1614182	1.24E-06
	gnomAD	2.1.1	11-57003976-C-G(GRCh37)	p.Arg168Pro	1	250698	3.99E-06
R168L	gnomAD	4.1.0	11-57236502-C-A(GRCh38)	p.Arg168Leu	1	1614182	6.20E-07
	gnomAD	2.1.1	11-57003976-C-A(GRCh37)	p.Arg168Leu	NA	NA	NA
R168C	gnomAD	4.1.0	11-57236503-G-A(GRCh38)	p.Arg168Cys	24	1614074	1.49E-05
	gnomAD	2.1.1	11-57003977-G-A(GRCh37)	p.Arg168Cys	4	250710	0.00001595

Table showing allele frequencies for the T89M and R168H variants in the gnomAD dataset.

We further recalculated the *in silico* consequence predictions and deleteriousness scores using the Combined Annotation Dependent Depletion (CADD) web service (version GRCh38-v1.7) to confirm that T89 and R168 are highly conserved at the DNA and protein level, and are predicted “deleterious” and “probably damaging” by SIFT and PolyPhen2, respectively, as previously described in the manuscript. The resulting combined CADD (PHRED) scores for both variants above 26 suggests pathogenic consequences (see Table below), as described previously in the manuscript.

T89M			R168H		
Chrom	11	11	Chrom	11	11
Pos	57236739	57236739	Pos	57236502	57236502
Ref	G	G	Ref	C	C
Alt	A	A	Alt	T	T
Type	SNV	SNV	Type	SNV	SNV
SIFTcat	NA	deleterious	SIFTcat	NA	deleterious
SIFTval	NA	0	SIFTval	NA	0.02
PolyPhen Cat	NA	probably_da maging	PolyPhen Cat	NA	probably_da maging
PolyPhen Val	NA	0.989	PolyPhen Val	NA	1
priPhCons	0.836	0.836	priPhCons	0.723	0.723
mamPhC ons	0.997	0.997	mamPhC ons	0.851	0.851
verPhCon s	1	1	verPhCon s	0.999	0.999
priPhyloP	0.595	0.595	priPhyloP	0.418	0.418
mamPhyl oP	4.41	4.41	mamPhyl oP	1.555	1.555
verPhyloP	7.994	7.994	verPhyloP	4.233	4.233
GerpRS	5159.84	5159.84	GerpRS	5159.84	5159.84
GerpRSpv al	0	0	GerpRSpv al	0	0
GerpN	6.4	6.4	GerpN	6.4	6.4
GerpS	6.4	6.4	GerpS	5.49	5.49
RawScore	5.010961	5.010961	RawScore	4.915762	4.915762
PHRED	28	28	PHRED	27.4	27.4

Table showing the key recalculated SIFT and PolyPhen-2 predictions of deleteriousness for the selected T89M and R168H variants, with resulting combined CADD (PHRED) scores for both variants above 26 suggests pathogenic consequences.

We have also performed further mathematical modelling using AlphaMissense to confirm the predictions of deleteriousness and, whilst this was ambiguous for T/M89 in this setting, R/H168 was indeed predicted to be likely pathogenic (see Figure below). These findings were also reproduced using BayesDel deleteriousness meta-scores.

Spot mutations

The database and web application was updated on 2024-05-06. If you encounter some issue, please, let us know.

Identifier:

APLNR (APJ_HUMAN, P35414, ENST00000611099.1)

a

a.a.	benign	ambiguous	pathogenic	mean
L84	2:M,V		3:P,Q,R	0.666
W85			5:C,G,L,R,S	0.983
A86	1:S	1:G	4:D,P,T,V	0.734
T87	3:A,I,S	1:N	1:P	0.338
Y88	1:F		5:C,D,H,N,S	0.735
T89	2:A,S	1:M	3:K,P,R	0.558
Y90	5:C,F,H,N,S	1:D		0.223
R91	4:G,L,Q,W		1:P	0.34
D92	5:A,E,G,H,N	1:Y	1:V	0.298
Y93	1:F	2:N,S	3:C,D,H	0.522
D94	5:A,E,H,N,Y	2:G,V		0.319
W95			5:C,G,L,R,S	0.993
P96		3:A,L,T	3:H,R,S	0.539

b

a.a.	benign	ambiguous	pathogenic	mean
A161	2:G,S		4:D,P,T,V	0.729
M162	4:I,L,T,V		2:K,R	0.376
P163			6:A,H,L,R,S,T	0.941
V164	2:A,I	1:G	3:D,F,L	0.528
M165	2:L,V	2:I,T	2:K,R	0.52
V166	3:A,L,M	1:G	1:E	0.345
L167	1:F	2:I,V	1:S	0.429
R168			6:C,G,H,L,P,S	0.964
T169	4:A,I,N,S		1:P	0.258
T170	2:A,S	3:I,N,P		0.298
G171	4:A,E,R,V	1:W		0.3
D172	7:A,E,G,H,N,V,Y			0.09
L173	5:F,M,S,V,W			0.141

Figure showing the AlphaMissense predictions for pathogenicity for (a) T89M and (b) R168H apelin receptor variants. The R168H variant was confirmed as likely to be pathogenic in this model.

We also re-analysed data from Uniprot and confirm that both T89 and R168 amino acid sites are highly evolutionarily conserved across species (see Figure below), as indicated in the manuscript, and supports the concept they are important for ligand-receptor interaction.

Protein identifiers

UniProt P35414
 InterPro IPR000276, IPR003904, IPR017452
 Pfam PF00001

3D Structures (PDB)

2LOT, 2LOU, 2LOV, 2LOW, 5VBL, 6KNM, 7SUS, 7W0L, 7W0M, 7W0N, 7W0O, 7W0P

Genomic identifiers

Ensembl ENSG00000134817
 UCSC uc058bkm.1
 RefSeq NM_005161
 NCBI 187
 HGNC 339

Search databases

- Expression Atlas
- GeneCards
- Protein Atlas
- Missense3D-DB
- Entries in DECIPHER for this gene

APLNR: P35414

Transcript used in protein view: ENST00000606794.2, NM_005161.6 1 exons, 380aa MANE Select

Figure confirming the high evolutionary conservation of the T89 and R168 amino acid sites (circled in red) across species in the UniProt database.

We thank the referee for the suggestion that a haplotype map of the apelin receptor gene would be useful to predict the probable occurrence and sites of variants. However, deciphering the genetic differences that make some people more susceptible to disease than others is beyond the scope of the current manuscript.

2. The observation that small molecule agonists bind deeper in the orthosteric pocket of the receptor than peptide ligands ELA and apelin is interesting. The R168 amino acid appeared as amino acid residues interacting with both peptides and small molecules in the list of Figure 6, but the R/H168 variant is not crucial for small molecule agonist to bind the receptor. It would be necessary to investigate which residues are important for small molecule-receptor binding. In addition, since ligand-receptor interaction is a matter in this study, the structure of ligands and small molecule agonists would be necessary to provide for better clarification of similarity and difference in structure of peptides and small molecules.

We thank the reviewer for highlighting this important point regarding the observed difference in small molecule vs peptide binding. For future drug development one strategy is to make peptide analogues but, as we report (Davenport et al., Nat Rev Drug Discov 2020. PMID: 32494050) this required modifications to improve PK/PD or the alternative is to develop small molecules. Therefore, this structural observation may be of interest for future drug design. We have clarified in the revision that we have reported residues important for small molecule-receptor binding in several figures and all of the relevant small molecule structures, CMF-019, AMG-986 and CMPD 644 have been disclosed in the manuscript and supplementary data file.

As we indicate in reply to Referee 1 (point 1 above), we have also re-analysed our X-ray structural data in the context of comparing the peptides and small molecule agonists in comparison with the recently published cryo-EM structure (PMID: 38428423). As explained above, this paper was published a week after we had submitted our manuscript and compliments our results from the X-ray structure.

Importantly, PMID: 38428423 show the same conclusions as one aspect of our manuscript; the amino acid residues in the binding pocket of the receptor interacting with the small molecule ligands. This comparison is included in new text in Fig 6 as well as new text in the results and discussion sections of our revised manuscript. A major part our manuscript is different from PMID: 38428423, as we have focussed on understanding the intriguing differences between the two endogenous peptides, apelin and ELA, binding to the receptor, informed by the discovery and characterization of naturally occurring apelin receptor variants for the 100,000 genomes. Naturally occurring apelin receptor variants in humans have not previously been described. As we indicate in our revised Supplementary Data file we were able to use the information to design and synthesise both G protein biased and balanced peptide apelin agonists (Figure S 10).

3. The T/M89 variant narrows binding pocket for ELA, resulting in reduced binding affinity to ELA-11, whereas R/H168 variant diminished both peptide binding to the receptor. Does T/M89 variant exert any phenotypes in humans/mice or in hESC-CM? If any pathological and physiological relevance of this variant should be investigated.

In the 100,000 genomes T/M89 variant is from the cohort with a broad classification of 'primary immune disorder' which incorporates over 400 broad clinical conditions. However, as indicated above further clinical relevance of the variants identified in this study would require extensive clinical characterization in the patient and is beyond the scope of this study. We attempted to use base editing to recapitulate T/M89 in cardiomyocytes but unlike the R/H168, it was not successful.

4. Does their proposed strategy contribute to development of better G-protein-biased agonist? Very recently, Wang, et al. reported that they developed a claiming better G-protein-biased agonist than CMF-019 by structure-based designing (Cell. 2024 Mar 14;187(6):1460-1475).

We thank the referee for this comment. Both G protein biased (over beta-arrestin) and non-biased peptide agonists have been designed and tested in functional assays.

In the figure below we show our data for over 50 peptide apelin agonists which have a range of affinities (micromolar to subnanomolar) for the apelin receptor, determined in clinically relevant human left ventricle in competition binding experiments with [¹²⁵I]-apelin. These were all demonstrated to be agonists in the Gi

coupled cAMP assay. However, over half of these novel peptides exhibit no activity in a beta-arrestin recruitment assay, indicating clear G protein bias.

Fig. S10: Pharmacological characterisation of novel G protein-biased and not biased apelin receptor peptide agonists. **a** Over 50 peptide apelin receptor ligands were designed and synthesised that demonstrated high affinity (pK_i) in competition binding assays using [¹²⁵I]apelin-13 and sections of human left ventricle (HLV) and were agonists (pD₂) in a G_i coupled cAMP assay in CHO-K1 cells expressing the human apelin receptor. Data also shown for Pyr¹]apelin-13 (●). **b** Using [Pyr¹]apelin-13 as the reference compound (●), novel peptides were either G protein biased (○) or not biased (○) agonists (pD₂) when tested in the G_i coupled cAMP assay and β-arrestin recruitment assay. **c** Table shows data for two novel peptides, one G protein biased and one that is not biased, both of which have half-life values of 1-4 hours compared to the very short *in vivo* half-life of [Pyr¹]apelin-13 in rat. pK_i is the -log₁₀ of the affinity (dissociation constant K_i measured in a competition binding experiment). pD₂ is the -log₁₀ of the concentration producing half-maximal response in the functional G protein and β-arrestin recruitment assays.

5. For the assessment of physiological relevance of apelin receptor signaling, they used hESC-CM and measured three parameters of cell differentiation efficiency, including TnT expression, apelin secretion and voltage sensing (Figure 8). However, it is important to measure cardiovascular functions in adult animals or isolated cells, such as blood pressure, cardiac hypertrophy, heart contractility in animals, ex vivo isolated blood vessel contractility or in vitro cultured cardiomyocyte hypertrophy. Cell differentiation include complicated process and signaling in addition to G protein signaling and beta-arrestin signaling. It is not suitable to measure the potency of drugs and nor to discriminate G protein signaling and beta-arrestin signaling.

We thank the referee for requesting additional information on the physiological relevance of apelin receptor signalling, we have previously carried extensive studies to measure cardiovascular functions in adult animals, such as blood pressure, cardiac hypertrophy, heart contractility in animals, and actions *in vitro* cultured vascular endothelial cells for apelin, ELA and CMF-019 as outlined below. In all these studies the actions of apelin and ELA are comparable. We have added a sentence in the introduction to signpost readers to references in which cardiovascular *in vitro* and *in vivo* studies on apelin and ELA peptides have been reported. Data from our some of our previously published studies are provided below.

1) *in vivo* Cardiovascular actions of apelin/ELA peptides in rodents:

From Yang et al., Circulation. 2017 PMID:28137936.

Acute administration of both ELA-32 and [Pyr¹]apelin-13 to anaesthetised rats dose-dependently, increased ejection fraction (LV and RV) as determined by MRI. Using cardiac catheterisation of the LV both apelin and ELA dose dependently increased contractility (dP/dtmax mmHg), stroke volume cardiac output and LV systolic pressure without significant effects on heart rate. Carotid catheterisation following 21 days of ELA treatment showed no change in blood pressure with chronic administration in rats. Both ELA and [Pyr¹]apelin-13 caused acute systemic vasodilatation (measured by carotid artery catheterisation), Chronic ELA treatment for 21 days did not significantly affect systemic blood pressure. The figure below shows that 21 Days of ELA administration (n=5) did not affect systolic blood pressure (SBP), diastolic blood pressure (DBP), mean arterial pressure (MAP) or left ventricular systolic pressure (LVSP) compared to saline control (n=5).

In the rat MCT model of PAH both apelin and ELA administration over 21 days show benefit with a reduction in RV hypertrophy and RVSP and decreased muscularisation of small pulmonary vessels.

2) Cardiovascular actions of small molecule biased agonist CMF-019:

From Read et al. Biochem Pharmacol. 2016. PMID: 27475715

In vitro human pharmacology and human apelin receptor cell based assays

The graphs show that CMF-019 competes for the specific binding of apelin to the apelin receptor in human, rat and mouse heart.

Fig. 4. Competition binding experiments in heart tissue homogenates. The specific binding of CMF-019 to human left ventricular homogenate (A, ●), rat whole heart homogenate (B, ■) and mouse whole heart homogenate (C, ▲).

Additionally, in cAMP assays CMF-019 demonstrated comparable potency to [Pyr¹]apelin-13 but was much less effective in β -arrestin recruitment and receptor internalisation assays. Data analysis generated bias factors of ~400 and ~5000 for the cAMP pathways vs β -arrestin recruitment and receptor internalisation pathways respectively for CMF-019 (Table below).

Table 1

Values of ΔLogR and relative effectiveness (RE) for CMF-019 compared to [Pyr¹]apelin-13 in cAMP, β -arrestin and internalisation assays.

	ΔLogR	RE
β -Arrestin	-1.98 ± 0.18	0.01
Internalisation	-3.14 ± 0.19	7.19×10^{-4}
cAMP	0.62 ± 0.16	4.19

ΔLogR is $\text{Log}_{10}(\tau/K_A)$ where τ is a measure of agonist efficacy and K_A a measure of functional affinity [35]. n values for each of the assays are as indicated in Fig. 5.

Table 2

$\Delta\Delta\text{LogR}$ and bias factors (BF, in bold) for CMF-019 compared [Pyr¹]apelin-13 in cAMP, β -arrestin and internalisation assays.

$\Delta\Delta\text{LogR}$ BF	β -Arrestin	Internalisation
β -Arrestin	n/a	1.17 ± 0.26 15
cAMP	2.60 ± 0.24 398	3.77 ± 0.25 5828

$\Delta\Delta\text{LogR}$ is the difference between ΔLogR values in the different pathways [35].

From Read et al., Front Pharmacol. 2021 PMID: 33716722:

CMF-019 decreased apoptosis in human pulmonary arterial endothelial cells (PAECs), a driver of early disease phase. CMF-019 was able to partially prevent TNF α /CHX induced apoptosis in human PAECs. Read et al.,;11:588669. doi: 10.3389/fphar.2020.588669.

3) *in vivo* Cardiovascular actions of CMF-019 small molecule in rodents:

From Read et al. Biochem Pharmacol 2016. PMID: 27475715

CMF-019 *in vivo*. Caused a dose dependent increases in left ventricular contractility in anaesthetised rats to (A, graph below) intravenous CMF-019 potassium salt (\blacktriangle , n=7–9) compared to saline (\square , n=3-5)

CMF-019, like [Pyr¹]apelin-13, increased stroke volume and cardiac output in anaesthetised rats.

CMF-019 reduced peripheral arterial pressure in anaesthetised rat *in vivo*.

(A, graph below) intravenous CMF-019 potassium salt (■, n 8) compared to saline (□, n 4). Dose-dependent effect blunted at highest concentration likely because of poor drug-like properties of CMF-019.

Minor comments

Some of the reference does not match the content of text. There seem errors in numbering the reference.

We thank the referee for this comment. We have amended the text to correctly cite Reference 51 (Macrae et al., 2021), and have added Wang et al., 2024 (Reference 33), which outlines the recent cryo-EM apelin receptor structure published after our initial submission. We have provided further in-text citations for References 7 and 8 (Chng et al., 2013 and Pauli et al., 2014) in the Introduction to provide further support for the role of the second endogenous apelin receptor ligand, ELA. We are happy that in-text citations now correspond with the appropriate references. The final reference count is 70, which complies with the Nature Communications Author Guidelines.

Reviewer #3 (Remarks to the Author):

This study by Williams et al. is a very well designed, written and executed study. The findings are of great interest to the community, as it provides a platform of using genetic variation to fast-track the identification and characterization of key sites that regulate ligand-receptor engagement. This could have a great impact on drug designing. Genetic variation in GPCRs may contribute to the differentiation of an individual's response to endogenous ligands, thereby giving rise to pathophysiological conditions and potentially affecting the efficacy of therapeutic agents. The authors characterized coding variants of the apelin receptor by leveraging in vitro pharmacology, molecular modelling, StaR, and gene editing in stem cell-derived cardiomyocytes technologies. They found T892.64 is an important residue in ELA binding, and R1684.64 is a forming extensive interaction with C-termini of both peptides. They also presented a novel apelin receptor crystal structure bound to the G protein-biased agonist CMF-019. Here are some of my comments.

1. The authors give different binding responses of receptor variants to synthetic molecules and peptides; how do these variants affect the clinical efficacy of treatments based on such agonists? Can this study provide a stratified analysis linking variant-specific receptor behaviour to potential clinical outcomes?

We thank the referee for this comment. Our previous research has been to identify peptide analogues of apelin that are biased toward the G protein pathway with reduced internalization and therefore desensitization of the receptor. We anticipate such agonists could be more efficacious in agonist activity particularly combined with longer plasma half-lives, despite a reduced binding of the endogenous peptide, apelin or ELA. As indicated above, we have already identified peptide compounds with G bias retaining high affinity, the predicted desired pharmacological profile. Stratified analysis linking variant-specific receptor behaviour to potential clinical outcomes could be of value to identify such patients.

Hauser et al. (PMID: 2924936) have reported on natural genetic variation in GPCRs (including functional regions such as drug binding sites) and confirmed experimentally these caused reduced responses to

approved medicines. We can speculate that based on knowledge of ligand receptor interactions from X-ray crystallography and these naturally occurring variants, compounds could be identified that would be effective in most patients, to overcome this natural variation.

2. The study notes a significant difference in the binding mode of CMF-019 compared to peptide agonists. In what ways does the binding mode of CMF-019 differ from that of natural peptide agonists, and what implications might these differences have for its function as an agonist?

As indicated in response to Referee 1, Point 3, the results of the new analysis are added to the text and modified figure 6 plus a new Supplementary Figure S4. The results of this new analysis in this context are added to the text.

3. The legends of Figures 1b and c do not correspond to the positions in the figures.

We thank the reviewer, amended as requested.

REVIEWER COMMENTS

Reviewer #1 (Remarks to the Author):

The authors have addressed my concerns, and I agree with the publication.

We are very pleased that the reviewer agrees that we have addressed their concerns in our detailed response to referees, the revised manuscript and supplementary file and now agrees with publication.

Reviewer #3 (Remarks to the Author):

The authors have addressed all of my comments.

We are very pleased that the reviewer agrees with referee 1 in that we have addressed their concerns in our detailed response to referees, the revised manuscript and supplementary file.

Reviewer #2 (Remarks to the Author):

In this revision, the authors toned down the fast-track identification of GPCR variants in the large-scale genome data. Instead, they put more focus on the significance of T89 and R168 residues in apelin receptor regulating receptor-ligand engagement. They repeatedly claim that in the manuscript text and rebuttal letter.

Point 1. Does their proposed strategy contribute to development of better G-protein-biased agonist? Very recently, Wang, et al. reported that they developed a claiming better G-protein-biased agonist than CMF-019 by structure-based designing (Cell. 2024 Mar 14;187(6):1460-1475, PMID: 38428423)

In response to this reviewer's previous comments above, they show the data that half of over 50 peptide apelin agonists are G-protein-biased agonist (Figure S10). The data should be shown in main figures with each peptide sequence and functional validation. The PMID: 38428423 showed that the identified small compounds are much superior to CMF-019 in its potency to improve cardiovascular function. If they say that they identified key sites regulating receptor-ligand engagement, do they develop any improved compounds or peptide ligands better than CMF-019 based on their findings?

We thank the reviewer for these further comments. Yes, to confirm we have compounds that are better than CMF-019. However, we would highlight again that this was not the purpose of the manuscript but in fact in our previous revision, we added data to the supplementary file showing that we have generated compounds which are more G protein biased (over beta-arrestin) than CMF-019 (Supplementary Figure 10B). We have in this revision included the individual data for the compounds used to generate Supplementary Figure 10.

Our previously published data using the same assays as used in this manuscript (Read et al., 2016, PMID:27475715) showed that CMF-019 was biased towards the G protein pathway over beta-arrestin) by three orders of magnitude (2250). In this second revision of the manuscript, we report compounds that are up to four orders of magnitude (for example, NXE'779, 76 200 fold selective for cAMP versus beta-arrestin) in a new Table S5. Please note that Sosei Heptares has now changed to Nxera.Life. The prefix for compounds was 'HTL' and now changed to 'NXE', which has been amended in the submitted files. Table S5 includes competition binding data for our compounds tested against [¹²⁵I]-apelin in clinically relevant human heart, and comparison of the potency of compounds in the G protein pathway (cAMP) versus beta-arrestin assays to show G protein selectivity. We have also added additional in vivo data from two representative compounds, one biased towards the G protein pathway over beta-arrestin (NEX'065) and one that displays no bias (NEX'515). As previously stated, these data importantly show that both non-biased and G protein-biased agonists produce the same desired short term beneficial actions: increasing cardiac output and decreasing in left ventricular systolic pressure and arterial pressure. This profile of a G protein-biased agonist has also been shown by us in proof of principle studies to be beneficial in the Sugden-hypoxia animal model of pulmonary arterial hypertension.

The main manuscript is already at the maximum of 10 display items (tables and figures); therefore the new Table S5 and Figures S11 have been added to the Supplementary file.

Point 2. In Figure 8, the authors tried to show the functional relevance of apelin receptor variants, this reviewer again points out that the cell differentiation assay of hESC-CM is not suitable for that purpose and that they are still failing to do so. Once again, the authors are advised to measure cardiovascular functions in adult animals or isolated cells, such as blood pressure, cardiac hypertrophy, heart contractility in animals, ex vivo isolated blood vessel contractility or in vitro cultured cardiomyocyte hypertrophy.

We thank the referee for this advice. Naturally occurring variants in the apelin receptor in humans have not been characterised before and the impact of a single amino acid change on functional responses is difficult to predict in G protein coupled receptors. We therefore used CRISPR base editing to introduce the apelin receptor variant into human stem cell derived cardiomyocytes. This is a technically challenging experimental technique but important to carry out in human cells for clinical relevance and as proof of principle that the variant may (may not) affect function.

These cardiomyocytes were used in as an initial screen and we were able to show potential functional significance of the variant in several assays (not just cell differentiation), including voltage sensing studies that showed that the R/H1684.64 hESC-CMs exhibited a prolonged time-to-peak voltage. This suggests an influence on the cardiac action potential in cardiomyocytes. Phase 0 of the cardiac action potential describes the rapid depolarization of the cell membrane voltage, predominantly relating to activation of Na⁺ channels and influx of Na⁺ into the cell. We can speculate for example, that apelin may influence activation of these ion channels. Genetic disorders of ion channels may present with familial forms of arrhythmia, for example SCN5A in Brugada syndrome and some forms of long QT syndrome.

As Referee 2 will appreciate, the request for in vivo data characterising animals, where base editing has been used to introduce this variant, is even more technically challenging and would take many months to generate. We are not able to provide carry out the studies suggested. However, we hope that publication of the manuscript will stimulate further research as to whether apelin receptor variants will have clinical consequences and inform the potential of the apelin signalling pathway in as a target in cardiovascular disease.

REVIEWER COMMENTS

We note Referees 1 and 3 have no further comments having previously confirmed we had addressed their concerns and agreed with publication. We thank Reviewer 2 for the further suggestions which are shown in purple text, with our responses for clarification in blue text in italics. Modifications to the main manuscript are shown as blue text, original text in black.

Reviewer #2 (Remarks to the Author):

[Comments] The authors provided the new Table S5 and Figures S11. They should mention the compound NXE'779 in the main text, which is 76,200 fold selective for cAMP versus beta-arrestin and is likely superior to CMF-019, as listed in a new Table S5. This is important, because they eventually claim the 'clinical relevance' of this study, and then readers always ask the potency of identified compounds. In addition, the context of NEX'065 and NEX'515 should be described in the discussion section. This would help general readers to understand this study better

We thank the referee for these suggestions and have added new text or made all the changes as requested. New and modified text added to the main manuscript:

p20

Identification of G protein peptide apelin agonists

We have designed peptide apelin agonists and pharmacologically characterised them to measure apelin receptor affinity and identify those with marked signalling pathway preference. Affinity of these peptides, determined in the human heart binding assay, were in the sub-nanomolar to high nanomolar range (Table S5). Twenty-one compounds were agonists in both a cAMP and a β -arrestin recruitment assay indicating lack of pathway preference, at least under *in vitro* conditions using expressed human receptors. In contrast an additional twenty-three peptides were agonists in the cAMP assay with potencies ranging from 1.25nM-100nM (Table S5) but lacked agonist activity in the β -arrestin assay at concentration up to 10-100 μ M. One of these compounds, NXE'779 was over 75,000 times selective for cAMP versus β -arrestin pathways which is an improvement on the pathway selectivity profile for CMF-019 in these assays³⁰. Two of these compounds, NXE'515 (no pathway selectivity) and NXE'065 (highly cAMP versus β -arrestin selective) were subsequently shown to have comparable actions in anaesthetised rats following bolus intravenous administration (30-150 μ g/kg). Both compounds significantly decreased arterial pressure, left ventricular systolic pressure and cardiac output to a similar extent (Fig S11) confirming what was reported for CMF-019²⁴ and MM07¹⁵ that G protein biased agonists retain the beneficial cardiovascular actions of the endogenous apelin peptides¹⁰.

p21

GPCRs form the largest class of therapeutic drug target for currently approved medicines. Identifying the mechanisms of ligand engagement at GPCRs will lead to the discovery of new medicines targeting these highly tractable drug targets^{2,3,4} In this study we report a series of apelin peptides. Our data confirm that G protein biased apelin peptides retain the beneficial *in vivo* cardiovascular actions of the endogenous apelin peptides and we have identified NXE'065 which has greater cAMP versus β -arrestin pathway selectivity than CMF-019 (Fig. S10, Fig. S11, Table S5). Here, we have used a combinatorial approach, encompassing the identification and *in vitro* characterisation of naturally occurring rare receptor variants, modelling, and a novel crystal structure bound to a G protein-biased small molecule, to pinpoint key structural determinants of endogenous peptide and synthetic ligand binding and activity at the apelin receptor. The data provide new insights that might facilitate and guide structure-based drug design at this physiologically and pathophysiologically relevant GPCR.

p28

Pharmacological characterisation of peptide agonists

Affinities were determined as described above; see Radioligand Binding section. cAMP (cAMP Hunter™ eXpress AGTRL1 CHO-K1 GPCR Assay; 95-0147E2CP2M) and β -arrestin recruitment (PathHunter® eXpress AGTRL1 CHO-K1 β -Arrestin-1 GPCR Assay; 93-1050E2CP2M) assays were carried out in cells expressing the human apelin receptor according to the manufacturer's protocol to obtain values of pD₂. Cardiovascular actions of NXE'065 (30,100 and 150 μ g/kg) and NXE'515 (100 and 150 μ g/kg) in anaesthetised rat were determined as previously described for CMF-019³⁰. Animal experiments were performed in accordance with guidelines from the local ethics committee (University of Cambridge) and the Home Office (UK) under the Scientific Procedures Act (1986).

[Comments] If they claim the functional relevance of R/H168_4.64, only by showing the results of cell culture experiments of hESC-CMs, they should tone down the ‘clinically relevant’ cells/settings in the text to simply mentioning ‘human’ embryonic stem-cell derived cardiomyocytes. If they still want to say ‘clinically relevant’ cells, they should establish iPSC cells from the patients with R/H168_4.64 variant, differentiate it into cardiomyocytes, and compare it with control wild type cells. If the detrimental phenotype is reproducible with the patient cells, they should also conduct CRISPR base-editing the variant cells to correct R/H168_4.64 to wild type. In addition, the derivation of cardiomyocytes was conducted only from one strain of hESCs (WA09 from WiCell). If another hESC strain is used for the same experiment, would the results be reproducible?

We thank the referee for the suggestion that we tone down in the text that R/H168_4.64 is ‘clinically relevant’ as we are only able to show culture experiments of hESC-CMs and use ‘human’ embryonic stem-cell derived cardiomyocytes.

We have made these changes to remove ‘clinically relevant’ to the manuscript (see p1 abstract, p18 and p21 in blue) and substituted ‘human’.

Having made these changes we note the referee does not require further experiments (CRISPR base-editing of patient cells). We believe that this manuscript reports the first time the successful CRISPR base-editing (in this case R/H168_4.64) in the apelin receptor in stem cells based on searching Pubmed. Base editing is still a comparatively new and challenging technique. Our manuscript is a proof of concept, that we believe will stimulate further investigation. This could include, as the reviewer suggests, obtaining consent from the patient and ethical approval, supported by our base editing results, to obtain tissue samples, in order to generate iPSCs with the variant and correct to wild type. As the reviewer will appreciate, this will be difficult and take time to establish the lines, to generate iPSC-CMs and correct them to wild type. As we have previously stated, the focus of this paper is about the structural changes induced by the variants.

The referee asks would the results be reproducible? As the referee will be aware, base editing is a modified form of CRISPR/Cas9 technology that introduces precise single base point mutations but does not introduce double stranded break (DSBs), does not rely on homology-directed repair (HDR) or require donor templates, in contrast to classic CRISPR/Cas techniques (DOI:10.1038/s41586-019-1711-4). In genetic editing, DSBs can lead to generation of undesired genetic changes (eg translocations). Relying on HDR to induce point mutations is inefficient due to competing cellular genetic repair mechanisms such as non-homologous end joining.

Base editing offers a system that results in reduced off target effects and greatly increased editing efficiency (DOI: 10.1016/j.tibtech.2019.03.008). As a result it is not conventional in the field to repeat in another strain. A survey of recent literature published in 2023-2024, combining ‘CRISPR base editing’ with ‘human embryonic stem cell’ in Pubmed shows that CRISPR base editing is carried out only on a single strain, without the need for repeating in a second (see for example, doi.org/10.1016/j.scr.2024.103496, DOI: 10.1016/j.scr.2024.103425, doi.org/10.1016/j.scr.2024.103343, doi.org/10.1016/j.scr.2023.103119, 10.1016/j.scr.2021.102467).

Although we have not repeated the base editing in a different strain, as we state in the manuscript (page 21) the variant R/H168_4.64 closely phenocopied a previous model, where we genetically manipulated hESC-CMs to generate an inducible apelin receptor knockdown that also reduced ligand binding (reference 51 and see additional comments responding to the last referee’s point below.

Furthermore, the detrimental phenotype is so unclear. They describe that both development and GPCR signaling may be important. However, they should clarify whether the phenotype is due to development or GPCR signaling. The authors are strongly advised to conduct RNA sequence analysis and determine how differentiation is impaired. Since TnT positive-differentiated cardiomyocytes are markedly decreased by R/H168_4.64 variant, it is not unlikely that regulation of Na channel is altered in the signaling levels, thereby time-to-peak voltage prolonged. Rather, simply undifferentiated cells are the cause.

We thank the referee for these points. Firstly, we can confirm that the voltage measurements were made on cells that were selected as visibly contracting and therefore had undergone differentiation to cardiomyocytes and not areas of cells that did not beat. This suggests undifferentiated cells were not the cause.

We have amended the text to clarify this point as indicated above in the main text (p21) and in methods (p28):

p21

Furthermore, the R/H168^{4,64} variant induced a perturbed phenotype in the hESC-CMs. Differentiation efficiency was significantly reduced, and time-to-peak voltage in cells that were visibly contracting, was prolonged compared to wild-type, closely phenocopying a previously reported inducible apelin receptor knockdown system in the same cell model⁵¹.

However, time-to-peak voltage in cells that were visibly contracting and therefore differentiated, was prolonged compared to wild-type, closely phenocopying a previously reported inducible apelin receptor knockdown system in the same cell model⁵¹. Intriguingly, in this model⁵¹, the SCN5A, (the gene encoding the pore forming α -subunit of the voltage-dependent cardiac sodium channel NaV1.5) was the most significantly downregulated differentially expressed 'druggable' gene that could be experimentally studied. This is consistent with a role for apelin modulating cardiomyocyte sodium currents¹¹.

p28

Generated videos were loaded into a custom MATLAB (R2021a) code designed to extract values for time-to-peak and decay time in groups of cells selected as visibly contracting.

The purpose of measuring the effect of the R168H variant on hESC-CM voltage was simply to confirm, as we hypothesised from structural studies, whether (in addition to decreased ligand binding and reduced differentiation efficiency), GPCR signalling via the apelin receptor would alter cardiomyocyte contractility. As we report in the manuscript, the contraction pattern was dysregulated, and voltage signalling in R168H hESCs was prolonged. The rationale for these experiments was that we had previously shown (reference 51, see above) in an inducible apelin receptor knockdown model in hESC-CMs that also reduced ligand binding and reduced differentiation efficiency.

However, when we carried out RNASeq to compare wild type with apelin receptor knock down hESC-CM, we found only a limited number of genes that were significantly differentially expressed up or down. Most were associated with focal adhesion kinase (FAK), a key mediator in the regulation of cell-matrix adhesion and migration, suggesting a role in development. FAK KO mice displaying cardiac developmental defects (DOI: 10.1242/jcs.109.13.2989). We did not find changes in adhesion assays in comparing the variant with the wild-type and there are no drugs or tool compounds currently available, to target molecules related to adhesion to unravel this potential role of apelin receptor in development.

However, SCN5A, (the gene encoding the pore forming α -subunit of the voltage-dependent cardiac sodium channel NaV1.5) was the most significantly downregulated 'druggable' gene that could be experimentally studied. Crucially, it could be directly related to the role that apelin modulates cardiomyocyte sodium currents (see for example, doi.org/10.1016/j.bbrc.2007.04.017, doi: 10.1016/j.yjmcc.2009.12.011.). We found as expected prolonged time-to-peak voltage compared to wild-type. We agree with the referee that the most likely hypothesis is that regulation of Na⁺ channel is most likely also altered in this R168H hESCs variant. We have amended the manuscript as indicated above to incorporate the referees suggestion that the role of the Na channel is impaired.

REVIEWER COMMENTS

We note Referees 1 and 3 have no further comments having previously confirmed we had addressed their concerns and agreed with publication.

We thank Reviewer 2 for the further suggestion which are shown in purple text, with our responses for clarification in blue text in italics. Modifications to the main manuscript are shown as blue text, original text in black.

Reviewer #2 (Remarks to the Author):

The authors claim that the phenotype of R168H variant-knockin hESC-CM is the same as that of apelin receptor-knockdown hESC-CM in their previous report and thus that it's not necessary to confirm the phenotype-genotype relationship.

For clarification, we did not say 'R168H variant-knockin hESC-CM is the same as that of apelin receptor-knockdown hESC-CM'.

We said the variant was 'closely phenocopying' (p21) the apelin receptor-knockdown hESC-CM.

Furthermore, the R/H168^{4,64} variant induced a perturbed phenotype in the hESC-CMs. Differentiation efficiency was significantly reduced. However, time-to-peak voltage in cells that were visibly contracting and therefore differentiated, was prolonged compared to wild-type, **closely phenocopying** a previously reported inducible apelin receptor knockdown system in the same cell model⁵¹. Intriguingly, in this model⁵¹, the SCN5A, (the gene encoding the pore forming α -subunit of the voltage-dependent cardiac sodium channel NaV1.5) was the most significantly downregulated differentially expressed 'druggable' gene that could be experimentally studied. This is consistent with the **hypothesis for a role for apelin receptor signalling in modulating cardiomyocyte sodium currents**¹¹. The data highlight the potential of using genetic editing to generate 'disease-in-a-dish' models for the apelin receptor.

Are their apelin receptor-impaired hESC-CMs always the same? Is autocrine loop of apelin ligand – apelin receptor really necessary for maintaining SCN5A expression, instead of ligand-independent activation of apelin receptor? How about involvement of elabela?

When wild type hESCs are treated with an apelin-neutralizing antibody throughout differentiation to hESC-CM, do they recapitulate R168H variant-knockin or apelin receptor-knockdown?

This reviewer does not agree that they confirm the phenotype-genotype relationship for R168H variant.

We added in the discussion that the time to peak voltage that SCN5A may have a role as this was suggested originally by referee 2 in revision 3. As we do not have any further data suggested by the referee about the role of elabela or neutralising antibody, as directed by the editor, we have added the caveat as shown in blue text, that this is a hypothesis for a role for apelin receptor signalling in modulating cardiomyocyte sodium currents.